

# Effect of wind speed on the size distribution of biogenic gel particles in the sea surface microlayer: Insights from a wind wave channel experiment

**Cui-Ci Sun[1,2,3]**, **Martin Sperling[1]**,  **Anja Engel[1]**

[1] GEOMAR Helmholtz Centre for Ocean Research Kiel, 24105Kiel, Germany

[2] State Key Laboratory of Tropical Oceanography, South China Sea Institute of Oceanology, Chinese Academy of Sciences, 510301, Guangzhou, China

[3] Daya Bay Marine Biology Research Station, Chinese Academy of Sciences, 518000, Shenzhen, China,

*Correspondence to*: Anja Engel (aengel@geomar.de)

Running title: Variation of gel particles in the SML as a function of wind speed

**Key words**: wind speed, biogenic gel particle, size distribution, air-sea interface,





**Abstract**
Biogenic gels particles, such as transparent exopolymer particles (TEP) and Coomassie
stainable particles (CSP), are important components in the sea-surface microlayer (SML). The
accumulation of gel particles in the SML and their potential implications for gas exchange
and emission of primary organic aerosols have generated considerable research interest in
recent years. Changes in the particle-size distribution (PSD) can provide important
information for the understanding of physical and chemical processes involving gel particles,
such as aggregation, degradation or loss. So far, little is known regarding the influence of
wind speed on the size distribution of marine gel particles in the surface microlayer. Here, we
present results on the effect of different wind speeds on the PSD of TEP and CSP during a
wind wave channel experiment in the Aeolotron. Total area of TEP and CSP were
exponentially related to wind speed in the SML. At wind speeds $< 6$ ms$^{-1}$, a strong
accumulation of TEP and CSP occurred in the SML, decreasing at higher wind speed and
becoming depleted above wind speeds of 8 ms$^{-1}$. Wind speeds $> 8$ ms$^{-1}$ also significantly
altered the PSD slope of TEP in the 2-16 μm size range toward smaller size. Changes in
spectral slopes at wind speeds $> 8$ ms$^{-1}$ were more pronounced for TEP than for CSP
indicating a high aggregation potential for TEP in the SML, potentially enhancing the export
of TEP by aggregates settling out of the SML. Our experiment provided evidence for the
control of wind speed on the accumulation of biogenic gel particles and their PSD changes,
providing a useful insight into particle dynamics and biophysical processes at the interface
between air and sea.



## 1 Introduction

The sea–surface microlayer (SML) is the thin boundary layer (~50-100 µm) between the atmosphere and the ocean. It is central to a range of global biogeochemical and climate-related processes (Cunliffe et al., 2013). Due to the high variability of physical, biological, chemical, and photochemical interactions, SML properties often differ from the underlying waters (ULW). Previous field and laboratory studies have shown that marine organic gels can been riched in the SML, and highlighted the importance of microgels for increasing gelatinous biofilm formation (Galgani et al., 2014; Wurl and Holmes, 2008) and mediating vertical organic matter transport, either up to the atmosphere or down to the deep ocean. To date, mainly two kinds of gel particles have been widely studied in seawater: transparent exopolymer particles (TEP), which include acidic polysaccharides, and Coomassie stainable particles (CSP) that are protein-containing particles and can serve as a nitrogen source for bacteria and other organisms (Alldredge et al., 1993; Cisternas-Novoa et al., 2015; Long and Azam, 1996; Passow, 2002). TEP are sticky and can increase coagulation efficiencies of particles in seawater (Chow et al., 2015; Engel, 2000). TEP thereby can enhance particle aggregation rate in the ocean and therewith influence element cycling, including trace elements (Passow, 2002). Changes in TEP abundance and size distribution in SML might therefore also influence particle dynamics in the water column below. The formation of TEP represents an abiotic pathway of repartitioning dissolved organic carbon into particulate organic carbon (Engel et al., 2004). Compared to TEP, much less is known for processes controlling CSP formation in SML. It has been suggested that CSP and TEP are different particles, because of their distinct spatial and temporal distributions in the ocean (Cisternas-Novoa et al., 2015; Engel and Galgani, 2016). It has also been reported that enrichment in CSP in the microlayer was related to bacterial activity, implying that bacteria may play a





pivotal role in mediating processes at the air-sea interface by contributing exudates and
proteins  released through cell disruption (Galgani and Engel, 2013).
Gel particle have been suggested to play an important role in air-sea exchange processes.
Previous results showed that gel particles with a polysaccharidic composition ejected by
bubble bursting events may act as cloud condensation nuclei (CCN) in low-level clouds
regions (Leck and Bigg, 2005; Orellana et al., 2011; Russell et al., 2010). Also proteinaceous
gels and amino acids can be enriched in the SML and in sea-spray aerosols (SSA)
(Kuznetsova et al., 2005). Since gel particles with fractal scaling provide a relatively large
surface to volume ratio, they are assumed to act as a cover at the interface between air and sea,
potentially reducing molecular diffusion rates (Engel and Galgani, 2016). Thus, the
enrichment of organic matter, including gels, in the SML could modulate the air-sea gas
exchange at low and intermediate winds (Calleja et al., 2009; Engel and Galgani, 2015;
Mesarchaki et al., 2015; Wurl et al., 2016). The SML is expected to disrupt at higher wind
speed, but the threshold wind speed for organic matter enrichment in general, and for specific
components in particular, is largely unknown (Liss, 2005). Wind was determined as a
principal force that controls accumulation of particulate material in the SML and as the most
important variable controlling the air-sea exchange of gas and particles (Frew et al., 2004; Liu
and Dickhut, 1998; UNESCO, 1985). Natural slicks often occur at low wind speeds (<6 ms$^{-1}$)
typically having wider area coverage for longer time in coastal seas compared to the open-
ocean (Romano, 1996). Using different SML sampling methods, such as the teflon plate, glass
plate and Garret screen (Garrett and Duce, 1980), direct relationships between wind speed and
SML thickness have been determined. Yet, the influence of wind on SML thickness is not
clear; Liu and Dickhut (1998) observed a decrease with wind speed up to 5m s$^{-1}$, while
Falkowska (1999) determined an increase up to a  wind speed of 8 m s$^{-1}$, beyond which the
thickness of the SML began to decrease. TEP enrichment in the SML has been described to





be inversely related to wind speed greater than 5-6 ms$^{-1}$ (Engel and Galgani, 2016; Wurl et al.,
2009; Wurl et al., 2011). One explanation that has been proposed for the reduction of TEP
abundance in the SML is that at higher wind speed, aggregation of solid particles with TEP
result in aggregates becoming negatively buoyant and sinking out of the SML. For
proteinaceous gels, Engel and Galgani (2016) observed that their enrichment was not
inversely related to wind speed. Yet, an inverse relationship between the slope of the CSP size
distribution in the SML and wind speed was observed, indicating larger CSP in the SML at
low wind speed. In addition, the dynamics of gel particles in the SML were also affected by
the other mechanisms that depend on the wind and wave conditions. It is proposed that gel
particles formation within the SML by bubble scavenging of DOM in the upper water column
(Wurl et al., 2011), because more TEP precursors are lifted up the water-column. Moreover,
compression and dilatation of the SML due to capillary waves may increase the rate of
polymer collision, subsequently facilitating gel aggregation (Carlson, 1983).
Particle-size distribution (PSD) is a trait description of gel particles that relates to many
important processes. It has been demonstrated that marine heterotrophs feed on gel particles
within specific size ranges (Mari and Kiorboe, 1996). Bacterial colonization TEP varies as a
function of the size (Mari and Kiorboe, 1996; Passow, 2002). Thus, changes in the size
distribution of biogenic gel particles will likely alter food-web structure and dynamics in the
ocean and the SML. Gel PSD and its variation with biogeochemical and physical processes
generally reflect the information about coagulation, break-up, and dissolution as well as on
sources and sinks of gels particles, either moving upward into or sinking out of the SML.  In
addition, the abundance and size of marine gels in the SML and in subsurface waters may
determine their potential fate as CCN in the atmosphere (Orellana et al., 2011). Wurl et al.
(2011) provided a conceptual model for the production and fate of TEP in surface waters and
the underlying controlling mechanisms. However, due to the lack of observational data, we do



not understand well how the PSD of marine gel particles in the SML varies as a function of
wind speed and wave action. Knowledge of the characteristics of gel particles such as
abundance, total area and PSD in the SML, and how they relate to wind speed may improve
our understanding the marine primary organic aerosol-cloud feedback processes and
accurately estimate trace gas fluxes from the ocean to the atmosphere. Here, we assess the
dynamics of PSD of marine gels particles, i.e. TEP and CSP, in the SML in responses to
different wind speeds and bubbling. This study was conducted in 2014 with natural Atlantic
seawater at the 'Aeolotron' facility in Heidelberg, a large-scale annular wind-wave channel
that allows for full control of wind speed.



## 2 Methods

### 2.1 Experimental set up

Effects of different wind speeds on the size distribution of organic gel particles in the SML were studied during the Aeolotron experiment from 3$^{rd}$ to 28th November 2014. In total 20.000 L of North Atlantic seawater were collected by the research vessel FS Poseidon, including ~14000 L of high salinity water collected at 55 m at 64° 4,90' N   8° 2,03' E and ~ 8000 L collected at 5 m depth near the Island of Sylt in the German Bight, North Sea. The water was transferred to a clear tank container and transported to the Heidelberg Aeolotron facility. The 'Aeolotron' is the largest annular wind/wave facility in the world with a total height of 2.4 m, and an outer diameter of 10 m. The wind speed was measured by Pitot tube and anemometer. In order to compare the wind speed with measurements in the field and in other facilities, the facility-specific reference wind speed $U_{ref}$ is of little use. Instead, the friction velocity $U_\star$ which is separately determined and converted into the value $U_{10}$ as described in Bopp and Jähne (2014). $U_{10}$ is the wind speed which were measured in ten meters height on the open ocean if the same friction velocity $U_\star$ as in the Aeolotron is assumed. A total of 7 experiments were conducted, with stepwise increase in wind speeds (equivalent to $U_{10}$,) ranging from 1.371 to over 18.652 ms$^{-1}$as shown in Figure1 and Table 1.During some of the high wind speed conditions (Table 1), bubbles were generated in addition with a profiO$_2$ oxygen diffuser hose to simulate strong breaking waves with bubble entrainment and spray formation. About 54 meters of this tubing were installed and were operated with a pressure of around 900 mbar with normal air taken from the air space of the Aeolotron at a flow rate of around 100 L min$^{-1}$. In addition, on day 5, day 12 and day 23, one wind speed was arranged at about 18 ms$^{-1}$ with and without bubbling for about 2 hour, respectively. Seawater temperature over the course of the experiment was about 21 °C ± 1.



For two periods (days 9-16 and 20-26) light was switched on, and provided
Photosynthetically Active Photon Flux Density (PFD) at the water surface of about 115-120
$\mu mol\ m^{-2}\ s^{-1}$ over about 20 m of the tank perimeter, and 20 $\mu mol\ m^{-2}\ s^{-1}$ for the remaining 10
m.
Temporal changes in hetero- and autotrophic plankton and neuston abundance and in organic
matter during the experiment are described in more detail in Engel et al. (2017) and are
summarized here only briefly. Heterotrophic microorganisms dominated cell abundance and
biomass in the tank during the whole study. Two peaks of bacterial abundance in the SML
occurred on day 4 and on day 11, respectively. On day 20, a seed culture of *Emiliania huxleyi*
(cell density: 4.6 x $10^5$ cell $ml^{-1}$) was added followed by a biogenic SML from a previous
experiment on day21.
**2.2  Sampling**
SML samples were collected with a glass plate sampler, made of borosilicate glass with
dimensions of 500 mm (length)×250mm (width) ×5 mm (thickness) and with an effective
surface area of 2000 $cm^2$ (considering both sides). For each sample, the glass plate was
inserted into the water perpendicular to the surface and withdrawn at a controlled rate of ~20
cm $sec^{-1}$. The sample, retained on the glass because of surface tension, was removed by a
Teflon wiper, and for each sample the glass plate was dipped and wiped about twenty five
times. The exact number of dips and the volume collected were recorded. Samples were
collected into acid cleaned (HCl, 10%) and Milli-Q washed glass bottles. Prior to sampling,
both glass plate and wiper were rinsed with Milli-Q water, and intensively rinsed with
Aeolotron water in order to minimize their contamination with alien material. The first
millilitres of SML sample were used to rinse the bottles and then discarded. The bulk water

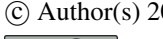



was sampled from the outlet at the middle-lower part of Aeolotron and collected into acid
cleaned (HCl, 10%) and Milli-Q washed glass bottles.
**2.3   Analytical methods**
Total area, particle numbers and equivalent spherical diameter ($d_p$) of gel particles were
determined by microscopy following Engel (2009). For TEP and CSP, 5 to 30 mL were
gently filtered (<150mbar) onto 25mm Nuclepore membrane filters (0.4 μm pore size,
Whatman Ltd.), stained with 1 ml Alcian Blue solution for polysaccharidic gels and 0.5mL
Coomassie Brilliant Blue G (CBBG) working solution for proteinaceous gels. The excessive
dye was removed by rinsing the filter with Milli-Q water. Blank filters for gel particles were
taken using Milli-Q water.
Filters were transferred onto Cytoclear © slides and stored at -20 °C until microscopy analysis.
For each filters, 30 images were randomly taken at ×200 magnification with a light microsope
Zeiss microscope (Zeiss AxioScope A.1). An image-analysis software (Image J, US National
Institutes of Health) was used to analyze particle numbers and area.
The size-frequency distribution of TEP and CSP gels was described by:

$$\frac{\mathrm{d}N}{\mathrm{d}(d_p)} = k d_p{}^{\delta} \tag{1}$$

where d$N$ is the number of particles per unit water volume in the size range $d_p$ to $(d_p+d(d_p))$
(Mari and Kiorboe, 1996). The factor $k$ is a constant that depends on the total number of
particles per volume, and $\delta$ ($\delta<0$) describes the spectral slope of the size distribution. The less
negative is $\delta$, the greater is the fraction of larger gels. Both $\delta$ and $k$ were derived from
regressions of log[$dN/d(dp)$] versus log[$dp$] over the size range 2–16 μm ESD. TEP carbon
concentration were calculated from the carbon-size relationship: *TEP-C*= 0.25 $r^{2.55}$ (pg C
TEP$^{-1}$), where *TEP-C* (pg C) is the carbon content of a given TEP particle with a radius $r$ (μm)



(Mari et al., 2001).
On day11, samples taken during wind condition 1 and 2 were contaminated and therefore
removed from data analysis.
**2.4 Data analysis**
Results from the SML samples were compared to those of bulk water and expressed as
enrichment factors (EF), defined as:
$EF = (C)_{SML}/(C)_B$ (2)
Where (C) is the concentration of a given parameter in the SML or bulk water, respectively
(GESAMP, 1995). Enrichment of a component is generally indicated by $EF > 1$, depletion by
$EF < 1$. Considering the measurement uncertainty of gel particles using microscopic method
within 10%, EF values $>1.1$ thus represent significant enrichment of gel particle in the SML,
while $EF < 0.9$ is determined to be a depletion. Enrichment or depletion was hence assumed
as being not unambiguously determinable for factors between 0.9 and 1.1.
Nonparametric statistics (Two Sample-Kolmogorov-Smirnov test) was performed to compare
differences of slope of gel particles size distribution between low and moderate wind speeds
($<8$ ms$^{-1}$) and high wind speeds ($>8$ ms$^{-1}$). In addition, statistical significance of changes
with respect to the slope of gel particles size distributionafter adding the seed culture of
*E.huxleyi* and the biogenic SML water from a previous experiment was determined with two
sample-Kolmogorov-Smirnov test on non-normalized anomalies given the data being normal
distributed. Average values are reported with ±1 standard deviation. Friedman ANOVA test
was carried out to evaluate bubble effect on enrichment in gel particles. Statistical
significance was accepted for $p < 0.05$. Calculations and statistical tests were conducted using
Microsoft Office Excel2010 and Origin 9.0 (Origin Lab Corporation, USA) software.



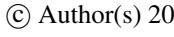

**3  Results**
**3.1  TEP and CSP developments in bulk and microlayer surface**
The developments of TEP and CSP in bulk and SML are shown in Figure 2. From day 1 to
day 15, average abundance and total area of TEP in the SML were $1.56\pm1.05\times10^8$ L$^{-1}$ and
$1.84\pm1.39\times10^3$ mm² L$^{-1}$, respectively. TEP abundance in the bulk water increased from the
initial $7.93\pm0.09\times10^7$ L$^{-1}$ on day 2 to $14.60\pm1.45\times10^7$ L$^{-1}$ on day 15. TEP total area in the
bulk water was $0.38\pm0.01\times10^3$ mm$^2$L$^{-1}$ initially and increased to the maximum value of 1.42
$\pm0.10\times10^3$ mm$^2$L$^{-1}$ on day 15. After addition of the *E.huxleyi* seed culture and of pre-
collected biogenic SML on day 20, TEP re-accumulated in the SML, with $2.15\pm0.72\times10^8$ L$^{-}$
$^1$ and $2.92\pm1.42\times10^3$ mm² L$^{-1}$, respectively.
Similar to TEP, CSP abundance and total area in the SML declined gradually between day 1
and 12. Here, CSP abundance in the SML decreased from $1.87\pm0.37\times10^8$ L$^{-1}$ to $0.30\pm0.16$
$\times10^8$ L$^{-1}$ on day 12, and CSP total area dropped from an initial $2.05\pm0.27\times10^3$ mm² L$^{-1}$ to
$0.156\pm0.065\times10^3$ mm$^2$L$^{-1}$.Generally, abundance and total area were less for CSP than for
TEP. The peaks of CSP abundance and total area in both SML and bulk water occurred on
day 24 corresponding to increasing of Chl*a* in the bulk water.
**3.2  TEP and CSP abundance and total area variations with respect to wind**

19       **speeds**

At the start of each wind experiment, the water surface was flat, without visible surface
movement. As the wind speed increased, the first capillary waves became visible and started
breaking above about $U_{10}$= 6 ms$^{-1}$ ($U_{10}$ =6.099 ms$^{-1}$ on day 22). At this wind speed, TEP



abundance in the SML decreased, except for day 15 and day 11, when TEP abundance
remained relatively stable or even increased slightly in SML at high wind speed (Fig. 3).
Similar to TEP, abundance and total area of CSP decreased with increasing wind speed,
excluding day 11 and day 2 (Fig. 4). The exponential decline of TEP and CSP total area in the
SML with increasing wind speed can be described by $\ln y = \ln a + b x$ or $y = a e^{bx}$, where x is wind
speed, $y$ is gel particle total area, $a$ is a constant depends on the total area of particles per
volume under the initial lowest wind speed condition. Exponential functions are the result of
constant relative growth or decline; here $b$ is the relative decline speed for the exponential
function. A measure of goodness of fit is the coefficient of determination (COD) denoted as $r^2$
($b_{\text{CSP-Totalarea}}$=-0.20±0.13, $r^2$=0.73±0.20, n=6; $b_{\text{TEP-Totalarea}}$=-0.18±0.07, $r^2$=0.87±0.19, $n$=5),
except for TEP area on day 11 and day 15, and CSP area on day 15. In contrast to total area,
only 3 out of 7 observations for TEP abundance and 2 out of 7 for CSP abundance were
exponentially related to wind speed. Thus, the relationship between abundance of gel particles
in the SML and wind speeds could not be well described by an exponential
function. Nevertheless, the reduction of gel particles abundance and area indicated a clear
removal from the SML with increasing wind speed. Enrichment of gel particles, with EF>1.2,
for both abundance and total area were generally found at wind speed 2-6 ms$^{-1}$ (Table 2),
excluding day 15 on which high CSP enrichment in the SML (EF's$_{\text{Abundance}}$=4.10 and EF's$_{\text{Total}}$
$_{\text{area}}$=3.20) was observed at wind speed of 18 ms$^{-1}$. Enrichment for both CSP and TEP was
higher for total area than for abundance at low wind speed (Table 2), suggesting selective
enrichment of larger gel particles in the SML.
**3.3 TEP and CSP size distributions related to wind speeds**
The power law relation fitted the gel particles size distribution ($d_p$:2-16 μm) very well for both
CSP and TEP under different wind speed conditions (mean of $r^2$=0.95) (Fig. 5 and Fig. 6).



Overall, no effect of varying wind speeds on the slopes of the whole size spectrum was
determined ($p > 0.05$; two-sample *Kolmogorov-Smirnov test*). However, a significant change
on TEP and CSP slopes was observed for wind speed >8 ms$^{-1}$ (Fig. 5 and Fig. 6 ) ($p < 0.05$;
two-sample *Kolmogorov-Smirnov test*). The slopes of size distributions for TEP ranged from -
2.93 to -1.32 (median of -2.17, $n$=17) at low and moderate wind speeds ($<8$ ms$^{-1}$) and were
significantly higher than those at high speeds ($> 8$ ms$^{-1}$) ranging from -4.05 to -2.48 (median
of -3.32, $n$=12) ($p < 0.05$; two-sample *Kolmogorov-Smirnov test*) (Fig. 7), excluding samples
in the SML collected from day 15. Moreover, 8 ms$^{-1}$was identified also as threshold below
which an obvious increase of maximal gel particle size in the SML was found except for day
15 (Fig. 8). Similar to TEP, the slopes of CSP were significantly smaller at high wind speed
($>8$ ms$^{-1}$) (-3.78 to -2.53, median of -3.10, $n$=12) than at < 8 ms$^{-1}$ (-3.21 to -2.16, median of -
2.59, $n$=17), again except for day15 ($p < 0.05$; two sample-*Kolmogorov-Smirnov test*) (Fig. 7).
However, during the second part of the experiment, when a seed culture of *E.huxleyi* was
added on day 20, followed by a biogenic SML from a previous experiment on day 21, no
significant difference of CSP size distribution was observed between high and low wind
speeds ($p$=0.06, two sample-*Kolmogorov-Smirnov test*), and the negative effect of increasing
wind on the maximum size for CSP was less obvious (Fig. 8). Compared to CSP, size
distribution of TEP was flatter at low and moderate wind speed ($p < 0.05$), and changes of
TEP spectral slopes related to wind speed were more pronounced (Fig. 7).
**3.4   Bubble effect on gel particles formation in the SML**
An effect of bubbling on the enrichment of gel particles in the SML was seen occasionally.
CSP were more enriched in the SML after bubbling in terms of abundance in 6 out of 7
experiments, albeit the EF's were only slightly above 1.2 (Table 3). In contrast to abundance,





enrichments for total area were less pronounced. Although no significant overall effect of
bubbling on SML enrichment in gel particles was determined, it should be noted that higher
enrichment factors were observed after bubbling for CSP on day 11 and for TEP on day 22
and day 24, respectively.

## 6  4   Discussion

### 7  4.1   TEP and CSP in SML related to wind speed

Driven by the wind, the friction at the water surface generates a shear current and turbulence
in the sea surface microlayer and thus the transport of gases and particles across the interface
are closely related. The complexity of the transport processes is caused by the wind blowing
over the surface, which not only causes a turbulent shear layer but also generates waves that
interact with the turbulent shear layer. The observed differences in concentration, enrichment
factor and PSD in response to changes in wind speed revealed that wind speed was a factor
controlling of gel accumulation in the SML during the Aeolotron experiment. Similar results
were observed during previous studies, which showed that TEP and particulate organic matter
concentrations in SML were negatively related to wind speed (Liu and Dickhut, 1998; Wurl et
al., 2011). Compression and dilatation of the SML due to capillary waves may increase the
rate of polymer collision, subsequently facilitating gel aggregation at lower wind speed (3-4
$ms^{-1}$) (Carlson, 1983). In addition, initial advection generated at wind speeds of 2-3 $ms^{-1}$,
maintain or enhance enrichments by increasing fluxes of potential microlayer materials to
surfaces (Van Vleet and Williams, 1983). As wind speed increases further (4-6  $ms^{-1}$), wave
breaking is likely to increase the turbulence in the top surface layer, but does not cause
homogenous mixing (Melville, 1996). This could lead to a reduced gel abundance in the SML,
but gel particle remain enrichment in the SML. For higher wind speeds of 8 $ms^{-1}$ and above,





the release of momentum from the wave breaking enhances the shear production of energy
and further increase the turbulence strength (Donelan, 2013). This could result in more break-
up of gel aggregates.
Selective loss of larger gels in SML was observed at high wind speed during the Aeolotron
experiment, suggesting stronger wind induced shear forces on larger gel particles in the SML.
This is because larger particles offer a greater surface area on which the stress of the fluid
shear is exerted (Spicer and Pratsinis, 1996; Zhang and Li, 2003). Large gels aggregates may
be weaker than smaller as porosity increases with size, and therefore the proportion of
constituent matter keeping the aggregate bound together decreases. Similar results were
achieved during previous empirical studies and numerical simulations of particle size
distribution (PSD) in marine waters, which demonstrated that aggregate breakage has
virtually no effect on the size distribution of small particles; instead strong shear of fluid may
selectively change the PSD for larger and fractal particles (Beauvais et al., 2006; Li et al.,
2004). When fractal scaling is incorporated for colloidal or solid particle characterization,
three linear regions with different slopes can also be identified in the size distribution, in
accordance with the three collision mechanisms, Brownian motion, fluid shear and
differential sedimentation (Kepkay, 1994). The process of collision of particle in shear is
typically important for particles larger than a few μm in diameter (Johnson and Kepkay, 1992;
Mccave, 1984; O'Melia and Tiller, 1993) and less important than Brownian motion for
particles in the submicron size range. It has been suggested that TEP can be formed
abiotically by coagulation of colloidal precursors with power law of gel particles size
distribution (Alldredge et al., 1993; Jiang and Logan, 1991). Slopes of the TEP size
distribution ranging from 2 to 16 μm in SML were in accordance with PSD slopes found
when fluid shear is dominant (Li et al., 2004). Also, wind speeds of 8 ms[-1] significantly
changed the PSD of TEP in size range of 2 to 16 μm, corresponding to the presence of wave

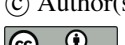



break and enhanced turbulence. Our observations suggest that PSD of TEP depends on shear
induced by wind and wave rather than Brownian motion or differential sedimentation.
Furthermore, the exponential decrease of total TEP area in the SML with increasing wind
speed may be related to the rate of turbulent kinetic energy dissipation $\varepsilon$ (cm$^2$ s$^{-3}$). The
relationship between gel concentration and turbulence has been reported to be of an
exponential form: $e^{(\varepsilon 1/2)}$ (Ruiz and Izquierdo, 1997). Therefore, increasing kinetic energy
dissipation, which is a linear combination of wind speed, wave and buoyancy forcing within
the mixed layer (Belcher et al., 2012), likely induces an exponential decrease in the total area
of gels in the SML. However, the exponential relationship was not observed between
abundance of gel particle and wind speed in this study, and no significant effect of wind speed
on PSD slopes was observed when including small gel particle ($dp$: 0.4-2 µm) into the
analysis. A likely explanation is that the abundance of gel particle was influenced not only by
turbulence levels, but also by bubble scavenging and bursting at higher speed. Particularly
smaller particles (submicron-micron size range), which contribute more to total gel abundance
rather than to total gel area, can accumulate in the SML due to bubble scavenging at high
wind speed. This may explain why changes in gel particles abundance did not fit well to an
exponential function with wind speed in our study.
It has also been suggested that CSP are less prone to aggregation than TEP (Engel and
Galgani, 2016; Prieto et al., 2002). During the Aeolotron experiment, the influence of wind
speed on spectral slopes was more pronounced for TEP than for CSP. In addition, size
distribution of CSP was not affected by wind speed, after adding the *E.huxleyi* seed culture.
Our results therefore support the idea that TEP are more prone to aggregation than CSP, and
potentially enhance the particle export of out of SML. Nevertheless, appearance of larger CSP
in the SML at low wind speed ($<$8 ms$^{-1}$) indicate that CSP are also involved in the formation
of surface slicks that becomes disrupted when wind speed increases.





## 4.2 Bubble effect on the enrichment of TEP and CSP

Prior studies have emphasized the importance of air bubbles for physical and biogeochemical

processes in the ocean upper layer (Kuhnhenn-Dauben et al., 2008; Thorpe et al., 1992). A

high gel particles load in the SML may be linked to upward transport by positive buoyant gel

particles (Azetsu-Scott and Passow, 2004), or to transport by rising bubbles (Wurl et al.,

2009). During this study, CSP and TEP abundance in the SML was more often enriched under

bubbling conditions than without bubbles. Proteins are known specifically for their surface

activity due to aliphatic groups rendering them intrinsic amphiphiles (Graham and Phillips,

1979). As a consequence proteins play a major role in the formation and stabilization of

bubbles (Dickinson, 2003). This may explain that CSP were more enriched in the SML after

bubbling during this study. Polysaccharide can interlink with protein by covalent bonding or

associate via physical interactions (e.g. by electrostatic and hydrophobic interactions, steric

exclusion, hydrogen bonding) and affect the interfacial characteristics of the fluid (Patino and

Pilosof, 2011). Sulphated polysaccharides interact with charged groups in a protein more

strongly than carboxylated hydrocolloids at pH above the protein isoelectric point (Dickinson,

2003). Therefore sulphated polysaccharides may be trapped by bubble-films also including

proteins (Mopper et al., 1995; Zhou et al., 1998), potentially leading to a higher enrichment of

sulfate half-ester groups in the TEP in the SML (Wurl and Holmes, 2008). Depending on the

hydrophobicity, different polysaccharide monomeric composition showed either competitive

or a cooperative behaviour with proteins (Baeza et al., 2005). Therefore, bubble enhancement

likely depends on the composition and proportion of polysaccharides and proteins within gel

particle during this study. Moreover, biological factors might affect bubble enhancement of

gel particles, i.e. on day 11, the SML was characterized by a strong enrichment of bacteria,

and on day 22 and day 24 autotrophic biomass increased (Engel, et al, 2017), corresponding

to the higher EF's$_{CSP\ abundance}$ on day 11 and EF's$_{TEP\ abundance}$ on day 22 and day 24, respectively,



under bubbling conditions. This observation is in accordance with findings by Zhou et al.
(1998) who showed that TEP formation induced by bubble was related to biological activity.
In addition, bubble size (Gantt et al., 2011; Oppo et al., 1999) is also an important factor
potentially determining the entrainment of organic matter in the SML. During the Aelotron
experiment, an aerator was used to simulate strong breaking waves for bubble entrainment
and spray formation. Unfortunately, the bubble size distribution was not determined. However,
under bubbling, the enrichment factors were higher for gel abundance than for total gel area,
indicating an increasing amount of smaller size gel particles (a few microns) in the SML. This
result is consistent with observations from the high Arctic, which showed that  short-chained
oligosaccharides might represent an important pool for the formation of small size particles
(microcolloids/particles) through bubbling (Gao et al., 2012).
**4.3   Implication of biogenic microgels in the SML**
In this study, values for $EF_{total\ area}$ of gel particles were higher than $EF_{abundance}$ at lower wind
speed. This suggests that large gel particles became selectively enriched in the SML and, due
to their larger surface area, may act as a cover sheet, potentially impacting processes across
the air-sea interface at low wind speeds. Based on the data of Aeolotron experiment, a
schematic diagram on interactions between physical dynamics and gel particle coverage in the
SML is proposed (Fig. 9). During this study, the enrichment of TEP and CSP in the SML
existed until wind speed reached 6 ms$^{-1}$, with strong enrichment at about 2-4 ms$^{-1}$, at which
slick streaks and bands were observed visually. Although surface tension measurements were
not made, values for the mean square slope, a measurement of surface roughness, were two or
three orders of magnitude higher at wind speeds > 6 ms$^{-1}$ than at wind speeds < 6 ms$^{-1}$ (Bopp
et al., 2017). The large total area of gel particle in the film may have contributed to the
observed reductions of wave slope, which would also imply corresponding reductions in mass



and momentum exchange at low and mediate wind speed (Frew et al., 2004). At wind speed
of 8 ms$^{-1}$ the sea surface became rougher, and micro-wave breaking started. In consequence,
the SML started to mix with the subsurface water leading to a more homogeneous distribution
of matter in the surface water column; thus a potential role of gel particles in gas-exchange
would be reduced. However, under conditions of high wind and wave breaking, microgel
precursors and nanogels can be aerosolized with sea spray (Gantt et al., 2011). For the ocean,
gel particle emission in aerosols has recently been discussed with respect to cloud formation,
precipitation, the hydrological cycle, and climate (Alpert et al., 2011; Knopf et al., 2011;
Wilson et al., 2015).

## 5   Conclusion

Our study showed that strong enrichment of biogenic gel particles in SML can occur at low
speed ($<$ 6 ms$^{-1}$) despite low autotrophic productivity in the water column. A negative
exponential relationship between the total area of gel particles in the SML and wind speed
was observed in most cases. Our results showed that the PSD slope is an important parameter
for characterizing the shape of the gel particle size distribution in the SML and reflects the
particles' fate in the SML (i.e. aggregation, fragmentation, sinking and injecting into air).
During the Aeolotron experiment, slopes of the TEP size distribution ranging from 2 to16 μm
in the SML were in accordance with PSD slopes of solid particles previously observed when
fluid shear is dominant. Moreover, the slope of PSD for TEP$_{(2-16\mu m)}$ and the maximum size of
gel particles varied significantly at about 8 ms$^{-1}$. Particle dynamics of TEP and CSP behaved
slightly differently. TEP appeared to be more prone to aggregation, potentially enhancing the
removal of particle out of SML. Responses of CSP enrichment to bubbling suggested that
proteinaceous particles are likely to be preferentially scavenged from the water column and



transported upward by bubbles. Overall, variations of gel particles sizes in the SML can
provide useful information on particle dynamics at the interface between air and sea.
To better understand the role of biogenic gel particles on biophysicochemical processes across
the air-sea interface, future studies should consider the full size spectrum of gels scaling from
nanometers to micrometers and also include their chemical composition. This could provide
important information on implications of marine gels for the aerosol and cloud formation as
well as for air-sea gas exchange.
**Data availability**
All data will become available at https://doi.pangaea.de/upon publication.

**Competing interest**
The authors declare that they have no conflict of interest.
**Acknowledgements**
We thank Tania Klüver, Ruth Flerus, Katja Laß, Sonja Endres and Jon Roa for technical
assistance. Armin Form helped to collect seawater for the Aeolotron experiment on board of
the *RV* Poseidon. This study was supported by the SOPRANIII project (03F06622.2) and by
China Scholarship Council (grant number 201408440016). We also thank Bernd Jähne,
Kerstin Krall and Maximilian Boppf or providing access and support during the Aeolotron
experiment, and for sharing their data and knowledge. This study is a contribution to the
international Surface Ocean Lower Atmosphere Study (SOLAS).





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



**Table caption**
Table 1.Wind speed settings on the Aeolotron regular experimental days
Table 2. The enrichment factor of gel particles abundance and total area at different wind
speeds
Table 3. Enrichment factors of TEP and CSP under Bubble bursting and without Bubbles
conditions



1  Table 1. Wind speed settings during the Aeolotron experimental days. ($^*$) indicates that an

2  aerator was used to simulate strong breaking waves with bubble entrainment and spray

3  formation under these wind speeds condition.

| Day | Wind velocity $U_{10}$ | | | | | | |
|---|---|---|---|---|---|---|---|
| 2 | NaN | NaN | 3.98 | 5.376 | 11.068 | NaN$^*$ | 17.892 |
| 4 | 2.092 | 3.437 | 4.305 | 8.306 | 14.145 | 13.448$^*$ | |
| 9 | 1.536 | 2.404 | 4.072 | 5.291 | 11.080 | 10.218$^*$ | |
| 11 | 1.663 | 2.89 | 3.925 | 8.033 | 13.955 | 18.208$^*$ | NaN |
| 15 | 2.580 | 4.985 | 6.424 | 11.095 | 18.090 | 17.991$^*$ | |
| 22 | 1.371 | 1.371 | 4.529 | 6.099 | 11.283 | 10.294$^*$ | 18.652 |
| 24 | 1.438 | 2.645 | 4.269 | 5.375 | 11.369 | 10.397$^*$ | 18.130 |



1    Table 2.Enrichment factors (EF) for gel particles abundance and total area in the SML at

2    different wind speeds (EF$_{Total\ area}$> EF $_{Abundance}$ are marked bold at low wind speeds)

| Experiment | Wind speed | TEP | | CSP | |
|---|---|---|---|---|---|
| day | (m s$^{-1}$) | EF$_{Abundance}$ | EF$_{Total\ area}$ | EF$_{Abundance}$ | EF$_{Total\ area}$ |
| 2 | **NaN(<4ms$^{-1}$)** | **2.24** | **7.40** | **41.43** | **113.98** |
| | 17.892 | 1.80 | 1.71 | 8.21 | 12.27 |
| 4 | **2.092** | **0.97** | **5.71** | **3.52** | **26.81** |
| 9 | **1.536** | **3.34** | **16.16** | **nd** | **nd** |
| | **2.404** | **4.80** | **12.76** | **1.84** | **7.08** |
| | **5.291** | **1.44** | **5.40** | **1.20** | **2.78** |
| | 11.080 | 1.08 | 1.07 | 0.74 | 0.72 |
| 11 | **3.925** | 0.91 | 1.16 | **13.46** | **31.16** |
| | 18.208 | 1.63 | 1.53 | 1.11 | 1.12 |
| 15 | 2.580 | 1.06 | 1.13 | 1.28 | 1.03 |
| | 4.985 | 0.48 | 0.77 | 0.47 | 1.08 |
| | 6.424 | 0.68 | 0.95 | 1.39 | 2.02 |
| | 11.095 | 0.77 | 0.70 | 2.14 | 1.50 |
| | 18.090 | 1.28 | 1.02 | 4.10 | 3.20 |
| 22 | **1.371** | **3.06** | **4.38** | **1.14** | **2.41** |
| | 4.529 | **3.06** | **5.04** | 5.46 | 4.54 |
| | 6.099 | **2.94** | **4.78** | **1.34** | **2.23** |
| | 11.283 | 0.61 | 1.02 | 1.07 | 1.02 |
| | 18.652 | 0.44 | 0.58 | 1.41 | 0.85 |
| 24 | **1.438** | **4.68** | **8.21** | **1.82** | **3.93** |
| | 4.269 | 6.97 | 6.06 | 5.94 | 6.82 |
| | 5.375 | 6.42 | 5.19 | 2.44 | 4.05 |
| | 11.369 | 2.38 | 1.23 | 0.72 | 0.66 |
| | 18.130 | 1.74 | 0.81 | 0.65 | 0.69 |



1   Table 3. Enrichment factors (EF) of TEP and CSP in the SML with and without bubbling of

2   the water column in the Aeolotron.

| Experiment day | Wind speed (ms⁻¹) | No bubbles | | Bubbles | | No bubbles | | Bubbles | |
|---|---|---|---|---|---|---|---|---|---|
| | | TEP Abund. | TEP Area | TEP Abund. | TEP Area | CSP Abund. | CSP Area | CSP Abund. | CSP Area |
| 4 | 13.448 | nd | nd | 3.11 | 1.91 | nd | nd | 1.47 | 1.12 |
| 9 | 10.218 | 1.08 | 1.07 | 1.29 | 1.24 | 0.74 | 0.72 | 1.70 | 1.12 |
| 11 | nd (~18) | 1.63 | 1.53 | 0.73 | 0.90 | 1.11 | 1.12 | 2.84 | 2.63 |
| 15 | 17.991 | 1.28 | 1.02 | 1.09 | 0.99 | 4.10 | 3.20 | 3.19 | 1.54 |
| 22 | 10.294 | 0.61 | 1.02 | 1.76 | 1.37 | nd | nd | 0.61 | 0.63 |
| 24 | 10.397 | 2.38 | 1.23 | 4.13 | 1.06 | 0.72 | 0.66 | 0.95 | 0.94 |
| 5 | 18.209 | 0.93 | 0.76 | 1.12 | 1.01 | 1.15 | 1.03 | 1.64 | 1.09 |
| 12 | 17.983 | 1.48 | 1.30 | 0.56 | 0.64 | 0.64 | 1.07 | 1.47 | 1.11 |
| 23 | 17.932 | 1.33 | 1.07 | 1.17 | 0.65 | 0.94 | 1.23 | 5.44 | 2.19 |





**Figure captions**
Figure 1: Schematic of wind speed ($U_{10}$) increase during the experiments conducted in the
Aeolotron.
Figure 2: Developments of TEP and CSP in the SML and the bulk water in the course of the
Aeolotron study.
Figure 3: Response of TEP abundance and total area to increasing wind speeds.
Figure 4: Response of CSP abundance and total area to increasing wind speeds.
Figure 5: PSD of TEP in the SML at different wind speeds. Linear regressions of
log(dN/d(dp)) vs. log(dp) were fitted to particles in the size range of 2-16 μm ESD, with wind
speeds $<$ 8 ms$^{-1}$ (solid line) and wind speeds $>$ 8ms$^{-1}$ (dash and dot).
Figure 6: PSD of CSP in the SML at different wind speeds. Linear regressions of
log(dN/d(dp)) vs. log(dp) were fitted to particles in the size range of 2-16 μm ESD, with wind
speeds $<$ 8 ms$^{-1}$ (solid line) and wind speeds $>$ 8 ms$^{-1}$ (dash and dot).
Figure 7: Box chart of slopes ($\delta$) of gel size distribution at low and moderate wind speeds ($<$
8 ms$^{-1}$) and at high wind speeds ($>$ 8ms$^{-1}$) (data for day 15 excluded due to no significant
response of PSD to wind speed on day 15)
Figure 8,A-D: Maximum size (ESD) of gel particles in the SML; A) and C): before addition
of *E.huxleyi*; B) and D): after addition of *E.huxleyi*.
Figure 9,A-G: A strong accumulation of TEP and CSP occurred at low wind speed as
determined by microscope, A: TEP (2.0 ms$^{-1}$), B: TEP (4.3 ms$^{-1}$), C: TEP (8.3 ms$^{-1}$), D: CSP
(2.0 ms$^{-1}$), E: CSP (4.3 ms$^{-1}$), F: CSP (8.3 ms$^{-1}$); G: Proposed schematic for interactions
between physical dynamics and gel particle coverage in the SML.



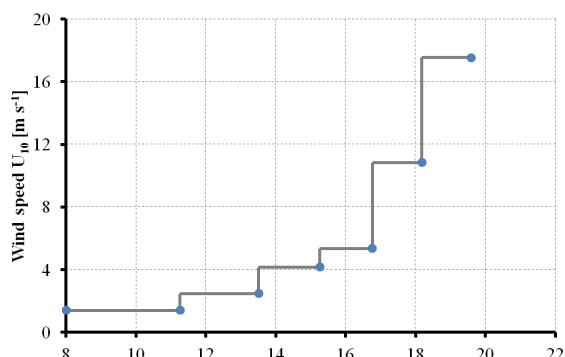

2                                    Figure 1



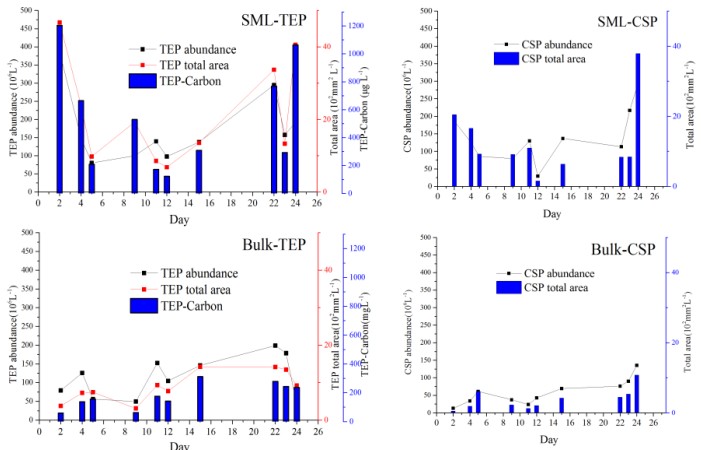

2                                  Figure 2



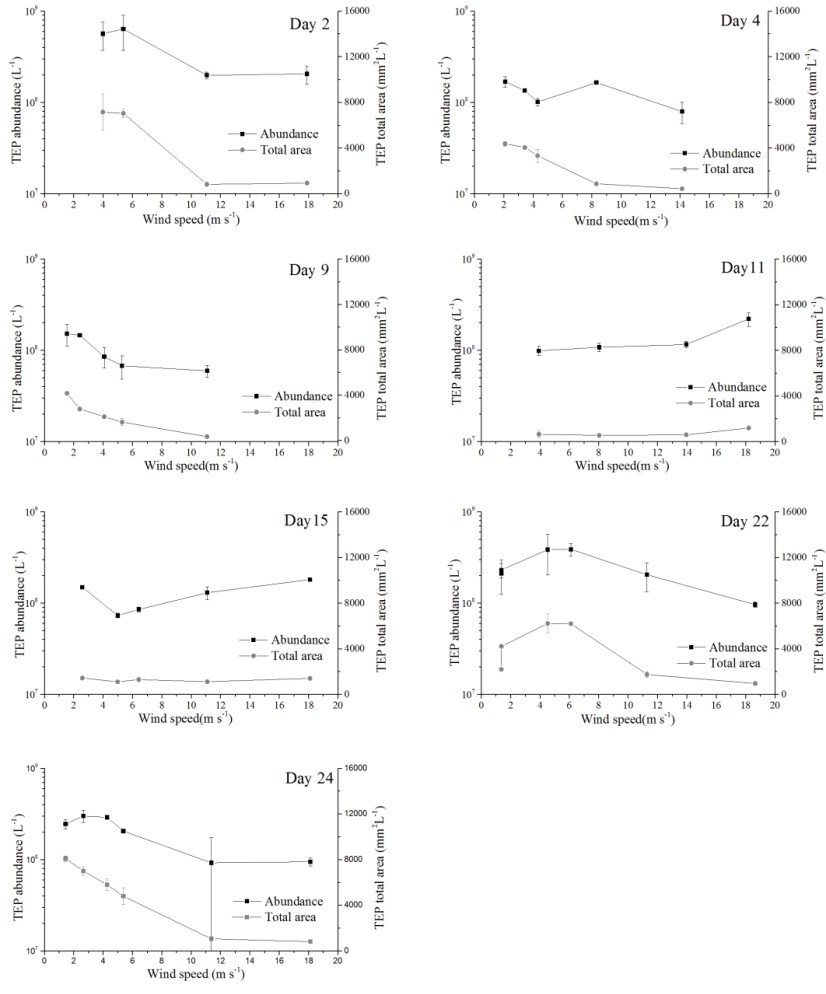

2                                            Figure 3





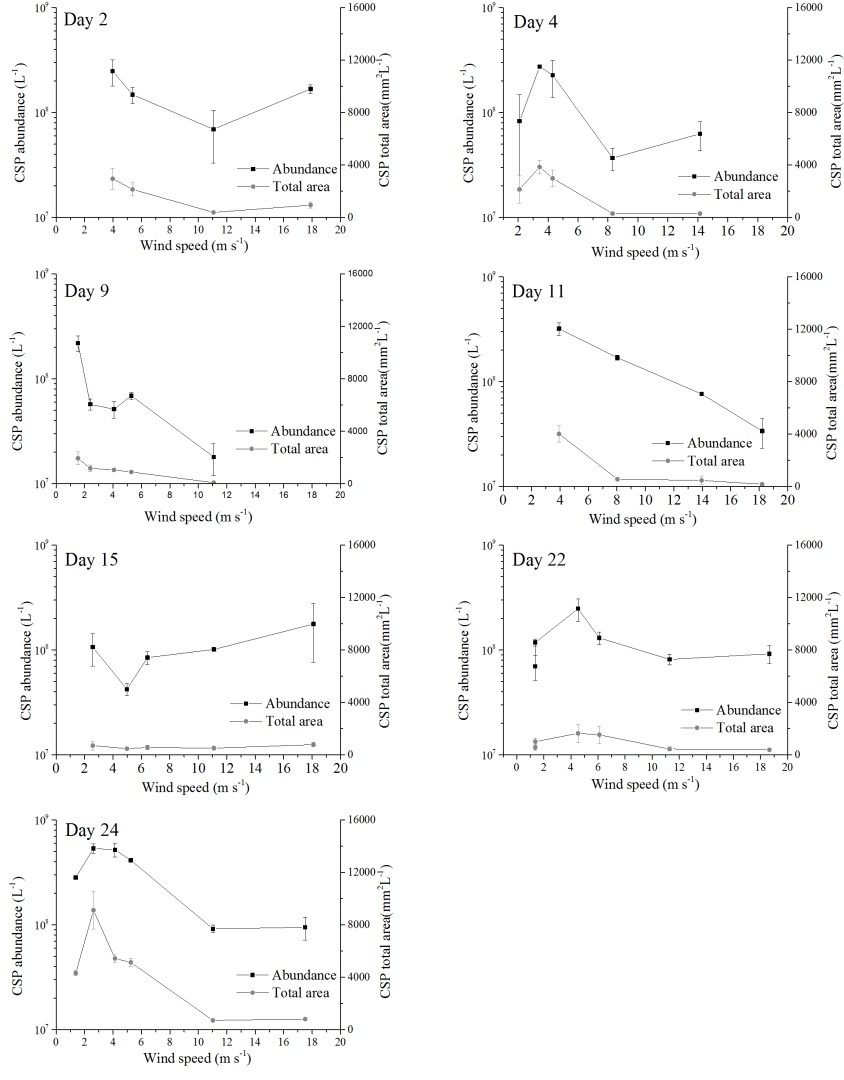

2                                                   Figure 4



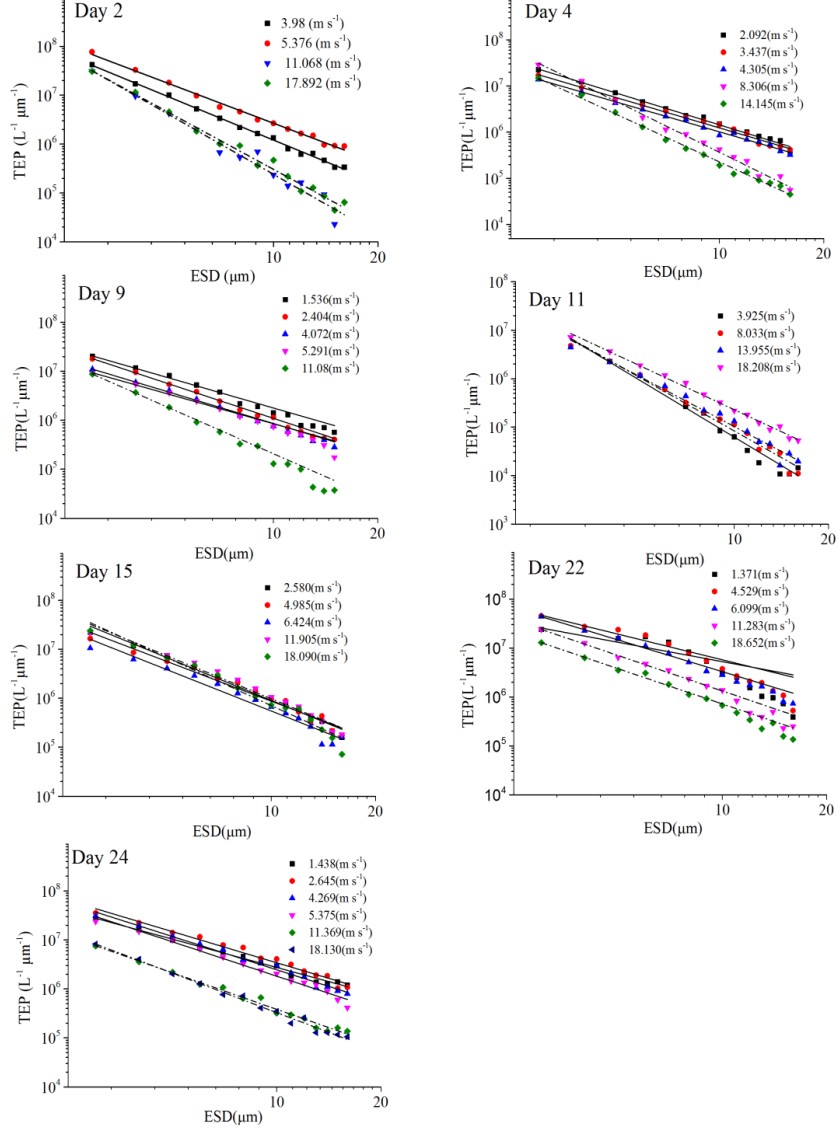

2                                          Figure 5



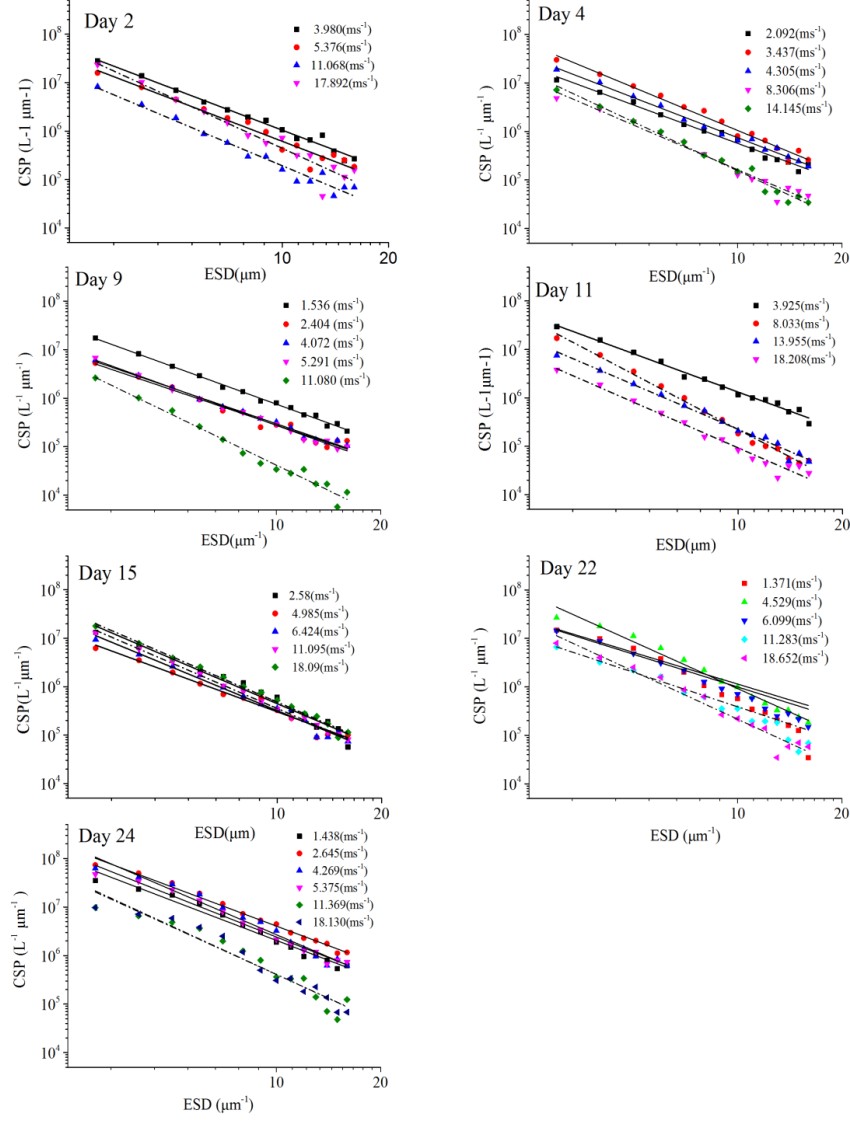

2                                              Figure 6





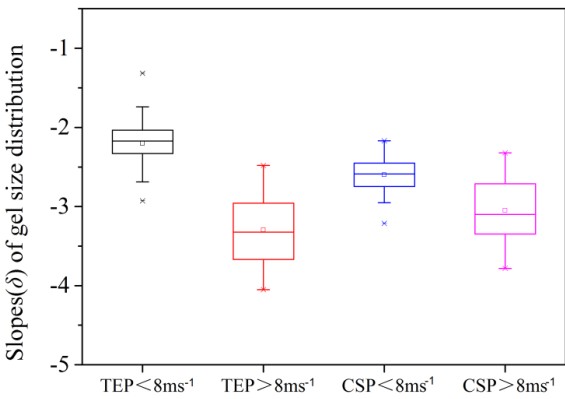

2          Figure 7





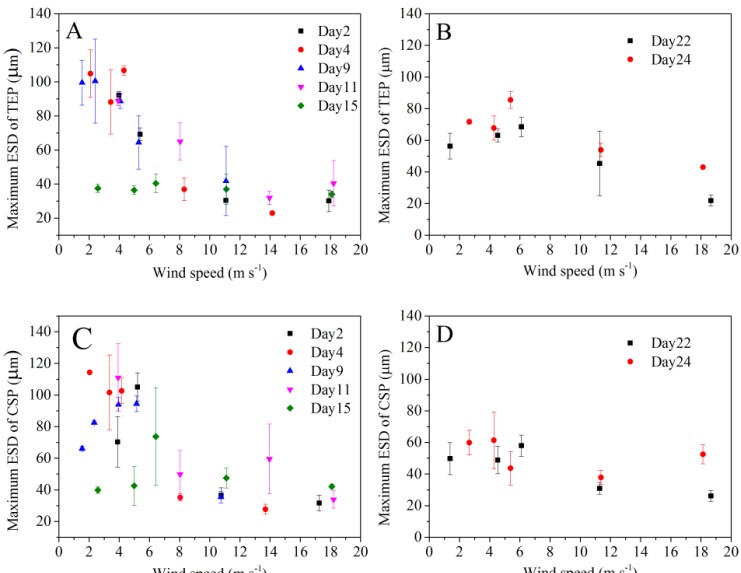

2                                    Figure 8



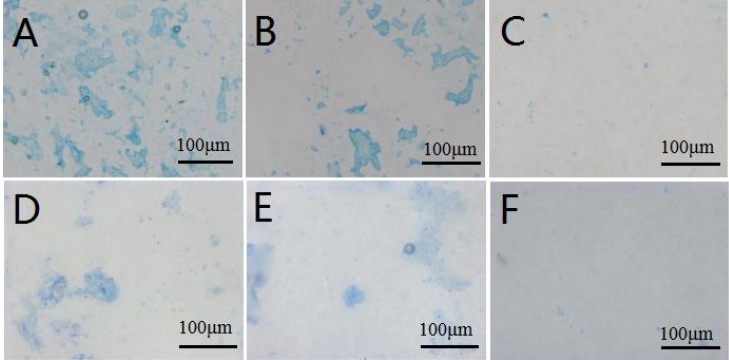

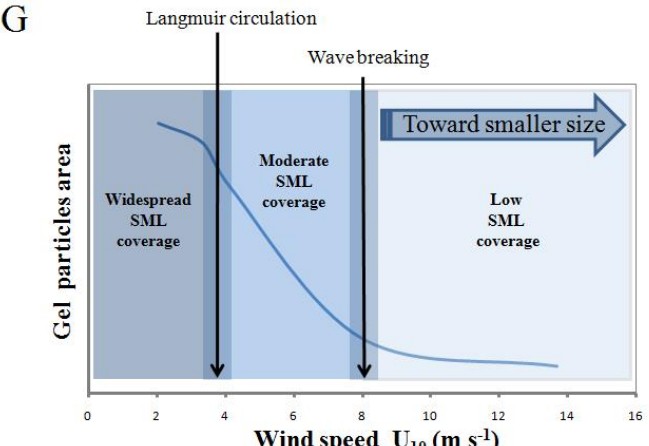

4                                Figure 9

