# Peer review of "Effect of wind speed on the size distribution of biogenic gel"

_Biogeosciences, 2017_

## Referee Comment (RC1) · Anonymous Referee #1 · 28 Nov 2017

Comment on bg-2017-419

This is an impressive experimental study that looks at the effect of increasing wind speeds on the accumulation of gel particles (transparent exopolymer particles and coomassie stained particles) and their size distributions in the surface microlayer (SML). Their main results point to an effect of high wind speed (>8 ms-1) altering the particle size distributions towards smaller gels, particularly for TEP, and to the disruption of the SML showing no enrichments in gels at those high wind speeds. The results deserve publication in BG; but in my opinion the way the results are presented and discussed can be improved. Specially the discussion section could be abbreviated and

focused better in their actual results. I provide some specific comments below.

Abstract:

Lines 2-9 I think this information can be abbreviated Line 15... and CSP? Complete the sentence

Introduction:

I think it is too long and repetitive. Maybe the intro could be abbreviated and re-organized as follows: (1) introduce the SML and its properties. (2) introduce gels, their PSD, their biochemical relevance, and their accumulation and role in the SML (3) role of wind speed in SML formation and in gel dynamics, particularly in PSD

Page 3 lines 6-7 can be enriched

Page 3 line 9: Start new line, since you talk about something different

Pages 3 line 6-8 and 4 lines 6-8 These sentences are repeating the same information.

M&M:

Page 7 lines 7-9, how long did it take from sampling to start the experiment?

Page 9 lines 22-23 I do not think the calculation of TEP-C is necessary in this study that does not focus on carbon fluxes

Page 10 line 17 include space between 'distribution' and 'after'

Results:

Page 11 lines 3-10 include average changes in TEP in the SML

Figure 2. Is this the average of the different wind speed conditions? Clarify. Include SD bars. I would use the same symbols for the same parameters; e.g., if columns are for total area (as they are in SML and bulk CSP), then use also columns for TEP total area. Anyway, I do not think it is necessary to show the TEP-C; as your paper is not

focused on these measurements. Include panel letters ABCD

Figure 3 and 4 say if this is SML or bulk water.

Figure 6. Day 22 panel: Use the same color and symbol code as in Figure 5 and in the rest of panels.

Page 11 lines 11-16. Here include average changes in CSP in bulk water

Page 11 lines 14-15 in bulk or in SML?

Section 3.3 Authors do not say whether they are describing PSD's in the SML or in bulk water at any moment. Assuming that this is only SML, some wording about changes in PSD in the bulk water could help understand these differences and to infer gel dynamics in the whole system through time

Page 12 lines 16-12 include some wording about enrichment factors in the high wind speed treatments

Page 12 lines 23-24 and page 13: Where, in the SML or in bulk? This differentiation should be clearly stated across the whole MS.

Page 13 lines 1-19 Maybe include the different slope values in a Table, as in the Figure it is hard to see if the difference is in slope or in the intercept

Discussion:

Page 14 lines 8-12 I don't think this sentence is necessary since you are not discussing any results.

Page 14 line 24 remain enriched

Page 15 line 7 gel aggregates

Page 15 line 4-page 16 line 2. This paragraph is very long and it is not clear how it is connected to the results obtained, which I think should be more carefully introduced in the discussion: For instance, do you refer to your measurements in the SML, in bulk

water, or in both? And, according to Kepkay 1994, shear is a dominant mechanism for particle aggregation; so how do you link this with the trend towards smaller gel particles at high wind speeds?

Page 16 lines 18-23. I do not see why. Average PSD are similar for TEP and CSP, and even lower for CSP at high wind speeds (page 13 lines 4-11). Or you said that because the change in PSD between high and low wind speeds was higher for TEP? Please clarify; and please refer to the results. To support this conclusion, maybe authors could look at the change in PSD of TEP and CSP through time; so check if these gel particles had been actually aggregated in the SML or not.

Section 4.3. I think it would be nice to comment about the changes in EF's through time. They apparently decrease until the phytoplankton culture is added (Table 2), even though you say that "a strong accumulation occurred in the SML (e.g. abstract line 13). How do you explain these decreases at low wind speeds?

I would appreciate some comments about your day 15; any explanation to this exceptional behavior?

End of review

---

## Referee Comment (RC2) · Anonymous Referee #2 · 19 Jan 2018

This paper presents results from wind-wave channel experiments on how wind-driven water mixing affects dynamics of marine gels (TEP and CSP) in the sea surface microlayer (SML). The authors conducted detailed analysis of TEP and CSP concentration and size distribution. They concluded that wind speed controlled gel accumulation and size distribution in the SML under their experimental conditions.

It is very difficult for me to evaluate the results and the conclusions in the present version of the manuscript because the description of the experiments as well as the presentation of some of the results are lacking important information (see details below). The manuscript would also benefit from shortening some of descriptive text in the

[Figure]

Intro and in the Discussion. See below for some suggestions on that as well. I strongly recommend that the authors edit the text so that it is more focused and less wordy.

Detailed comments:

Abstract: - In L. 9 use SML instead of surface microlayer. - Starting at L. 11: be more specific about the results on TEP and CSP; does this description refer to PSD of gels in the SML or bulk water? I suggest the following abbreviations for TEP and CSP in the SML (TEP-SML – CSP-SML) and in bulk water (TEP-bulk – CSP-bulk). Otherwise it is hard to distinguish between the two phases. - L. 17-18: You talk about the effects of TEP on aggregation and export. Since the focus of this paper is on TEP and CSP in the SML and the potential effects of gas exchange etc. you should focus/discuss potential effects on processes between the water and the atmosphere. In other words: if TEP settles out of the SML what could that mean for gas exchange processes between the water and the atmosphere.

Intro: Page 3 - L. 6: I don't think you need the abbreviation ULW. - L. 9: do you have a reference for this statement? - L. 14 -l. 2 on page 3 : In general, this text can be shortened as the focus is on SML sea-air exchange and not aggregation and particle export. Page 4 - L. 3: to me, your intro starts here. - L. 25 – l. 4 on page 5: In the first sentence you are saying that "TEP enrichment . . . is inversely related to wind speed . . .". You don't have to repeat this statement in the following sentence; the first part of that sentence can be shortened: "One explanation for this is that . . ..". - L. what are the "other mechanisms"

Methods: Page 7 - L. 4: change to "November 3-24, 2014." - L. 5: I am confused about the total volume of water collected for this study: Is it 20000 L with 14000 L of high sal water (what does high sal water mean??) + 8000 L at 5 m near Sylt? That does not add up, so remove "In total" in line 4, because your total is 42000 L. - L. 5: change to "were collected onboard FS Poseidon". How did you collect the water? Pumping or niskins? - L. 11-12: Info about something that you haven't used in your study like Uref

is not important, so delete this sentence. - L. 16: This is the part where I am getting confused about the experiments: 7 experiments were conducted, and you refer to fig. 1 and table 1 for explanation. Figure 1 shows the step wise increase of U which lets me believe that the 7 experiments were conducted under the same conditions of U. Table 1 leads me with a different impression as the values of U were quite different throughout the experiments (the table is lacking the unit for U; you also need to describe what 'NaN' means. Why are there no values for U at some days during experiment 7?). This needs to be explained in the methods. - L. 22-24: does this apply to all the 7 experiments? Page 8 - L.1-4: why was the light switched on in these two periods? Does that mean it was dark (0 umol m-2 s-1) throughout the rest of the incubation time? Why is this important? - L. 6: I could not find the Engel et al. 2017 reference in the list? Do you mean the Engel et al. (subm) reference? There is no way that we can get any information from this paper at this point. So you need delete this reference and give as many information of the methods as needed for this manuscript. - L. 7-8: This statement is too general, and I don't see why this would be important to know at this point. - L. 9-11: why was E. hux added to the water? I suggest adding some explanation in the intro. Also, what do you mean by "adding a biogenic SML from a previous experiment"? That is too vague, I have no idea what a biogenic SML could be/look like, and how can this be added without disruption etc. - L. 19: It would help to show the collection volumes or give a range because it is hard to imagine how much water you collected from the SML. Page 10: - L.2: what are the wind conditions 1 and 2?

Results: As mentioned above, I cannot evaluate the quality of the results before the authors improve the description of the experimental set-up. For example, I really cannot tell if the TEP and CSP results described on page 11 and shown in figure 2 are average values of all 7 experiments. Figure 2 also lacks error bars. You also need to add more detail to the figure legends (e.g. figs 4 and 5 show error bars, this needs to be mentioned in the legends). l. 16: this is the first time that chl a is mentioned. This needs to be described in the methods section. l. 20: what do you mean by "at the start

of each wind experiment"?? Does that mean that you varied the wind speed over a course of a day from 0 - 20 or so (see also figures 4 and 5). You lost me at this point . . ..

---

## Author Comment (AC1) · 29 Jan 2018

We much appreciate the referee's constructive and thoughtful comments. Below we have pasted in the entire review, and we have inserted our responses, indicated by two stars.

1. Abstract: Lines 2-9 I think this information can be abbreviated Line 15: : : and CSP? Complete the sentence. ** We agree and will edit here. The sentence 'The accumulation of gel particles in the SML and their potential implications for gas exchange and emission of primary organic aerosols have generated considerable research interest in recent years' will be deleted. The description on CSP will be added as well as the

information on PSD in the bulk water.

2. Introduction: I think it is too long and repetitive. Maybe the intro could be abbreviated and reorganized as follows: (1) introduce the SML and its properties. (2) introduce gels, their PSD, their biochemical relevance, and their accumulation and role in the SML (3) role of wind speed in SML formation and in gel dynamics, particularly in PSD ** We agree and will edit here according to the referee's suggestion.

Page 3 line 6-8 These sentences are repeating the same information. ** The repeating information will be deleted.

Page 3 line 9: Start new line, since you talk about something different ** Will be done.

3. M&M: Page 7 lines 7-9, how long did it take from sampling to start the experiment? **It took 41 days from sampling to start the experiment. The time of collection and starting experiment will be added: 'Effects of different wind speeds on the size distribution of organic gel particles in the SML were studied during the Aeolotron experiment from 3rd to 28th November 2014. In total 22000 L of North Atlantic seawater were collected on the 22. 09. 2014 by the research vessel FS Poseidon, including ∼14000 L of high salinity water collected at 55 m at 64° 4, 90' N 8° 2, 03' E and∼ 8000 L collected at 5 m depth near the Island of Sylt in the German Bight, North Sea.'

Page 9 lines 22-23 I do not think the calculation of TEP-C is necessary in this study that does not focus on carbon fluxes **We agree that this part can be removed.

Page 10 line 17 include space between 'distribution' and 'after' ** Will be adopted

4. Results: Page 11, lines 3-10 include average changes in TEP in the SML **We will add the average changes in TEP in the SML.

Figure 2. Is this the average of the different wind speed conditions? Clarify. Include SD bars. I would use the same symbols for the same parameters; e.g., if columns are for total area (as they are in SML and bulk CSP), then use also columns for TEP total area. Anyway, I do not think it is necessary to show the TEP-C; as your paper is not

focused on these measurements. Include panel letters ABCD. **The average of the different wind speed condition will be clarified in the revised version, and the SD bars will be added. The style of Figure 2 can be revised according to the suggestion, using the same symbols for the same parameters. TEP-C will be deleted.

Figure 3 and 4 say if this is SML or bulk water. ** Figure 3 and 4 refer to the SML. This information will be added in the revised manuscript and figure captions.

Figure 6. Day 22 panel: Use the same color and symbol code as in Figure 5 and in the rest of panels. **Will be adopted.

Page 11 lines 11-16. Here include average changes in CSP in bulk water **The information on average changes in CSP in bulk water will be added: 'Similar to TEP in the SML, CSP abundance and total area in the SML declined gradually between day 1 and 12. Here, CSP abundance in the SML decreased from $186.7\pm84.3\times106L-1$ on day 1 to $29.5\pm16.4\times106$ L-1on day 12. CSP total area in the SML dropped from an initial $20.5\pm2.7\times102mm^2$ L-1 to $15.6\pm0.7\times102mm2L-1$ on day 12. CSP concentration in the bulk water started with $12.9\pm10.7\times106$ L-1 in abundance and $0.5\pm0.04\times102mm^2$ L-1 in total area respectively, and increased to the first peak on day 9 for abundance and on day 5 for total area, and then declined.'

Page 11 lines 14-15 in bulk or in SML? ** We will add the information 'in SML'.

Section 3.3 Authors do not say whether they are describing PSD's in the SML or in bulk water at any moment. Assuming that this is only SML, some wording about changes in PSD in the bulk water could help understand these differences and to infer gel dynamics in the whole system through time **We agree that the assignment to the SML or to the bulk water is missing in the current version. We will thoroughly revise the text to give this information. Information of PSD's in bulk water will be added: 'Size distribution of gel particles (dp: 2-16$\mu$m) in the bulk water also followed the power law relationship of Eq. (2) (mean of r2=0.99), varying between -3.48 and -1.94 (mean value: -2.56, SD: 0.49) for TEP and between -3.43 and -2.01 (mean value: -2.50, SD: 0.42) for CSP.

For the slope of the size distribution in the bulk water, no significant difference was observed between high and low wind speeds. The PSD of both TEP and CSP in the bulk water and SML were shallower after adding a seed culture of E. huxleyi on day 20 and a biogenic SML from a previous experiment on day 21 (Fig. 8) (p<0.05, two sample-Kolmogorov-Smirnov test), i.e., the average slope of CSP in size of 2-16$\mu$m in the bulk was-2.84 before and -2.15 after addition of the E. huxleyi culture.

Page 12 lines 16-12 include some wording about enrichment factors in the high wind speed treatments **The information will be added.

Page 12 lines 23-24 and page 13: Where, in the SML or in bulk? This differentiation should be clearly stated across the whole MS. **It is in the SML. This differentiation will be clarified across the whole MS.

Page 13 lines 1-19 Maybe include the different slope values in a Table, as in the Figure it is hard to see if the difference is in slope or in the intercept. ** We agree and will add the table in the supplementary material.

4. Discussion: Page 14 lines 8-12 I don't think this sentence is necessary since you are not discussing any results. ** We agree and will delete it.

Page 14 line 24 remain enriched ** Will be adopted.

Page 15 line 4-page 16 line 2. This paragraph is very long and it is not clear how it is connected to the results obtained, which I think should be more carefully introduced in the discussion: For instance, do you refer to your measurements in the SML, in bulk water, or in both? And, according to Kepkay 1994, shear is a dominant mechanism for particle aggregation; so how do you link this with the trend towards smaller gel particles at high wind speeds? ** We will modify the paragraph to improve its understanding: 'Aggregation processes are primarily driven by encounter rates between particles that depend on particles concentration and turbulent shear (Mari and Robert, 2008; Ellis et al., 2004). It has been suggested that TEP volume concentration increases continuously under the low turbulence intensity by promoting the formation of TEP, but that TEP volume concentration and the fraction of large TEP are reduced at stronger shear (Mari and Robert, 2008). Thus, the effect of wind shear on gel aggregation is double-edged, and large aggregates may be broken apart when the turbulence intensity increases. In this study, the disruptive effect of the wind shear becomes more important when the wind speeds increases, therefore, larger gel became smaller at high wind speed. Reference: Ellis, K.M., Bowers, D.G., Jones, S.E.: A study of the temporal variability in particle size in a high energy regime. Coast. Shelf Sci. 61, 311–315,2004. Mari, X., Robert, M.: Metal induced variations of TEP sticking properties in the southwestern lagoon of New Caledonia, Marine Chemistry, 110, 98–108, 2008.

Page 16 lines 18-23. I do not see why. Average PSD are similar for TEP and CSP, and even lower for CSP at high wind speeds (page 13 lines 4-11). Or you said that because the change in PSD between high and low wind speeds was higher for TEP? Please clarify; and please refer to the results. To support this conclusion, maybe authors could look at the change in PSD of TEP and CSP through time; so check if these gel particles had been actually aggregated in the SML or not. ** According to the results, the average slopes showed about 41.2% changes at speed > 8 ms-1 for TEP in the SML, but only 23.8% for CSP in the SML. The change in PSD between high and low wind speeds was thus higher for TEP than CSP in the SML. In addition, after adding the E. huxleyi seed culture, no influence of wind speed on size distribution of CSP was detected. These results indicated that the influence of wind speed on PSD of gel particles in the SML may be more pronounced for TEP than for CSP, and that CSP are less prone to aggregation than TEP during the low wind speed.

Section 4.3. I think it would be nice to comment about the changes in EF's through time. They apparently decrease until the phytoplankton culture is added (Table 2), even though you say that "a strong accumulation occurred in the SML (e.g. abstract line 13). How do you explain these decreases at low wind speeds? I would appreciate some comments about your day 15; any explanation to this exceptional behavior? **We

think your suggestion is valuable. We will add the information: 'Pronounced changes through time in gel size slope and EF's were observed after the addition of E.huxleyi seed culture. At that time, shallower slopes for PSD of CSP and TEP revealed a higher abundance of larger gel particles relative to smaller ones for both SML and bulk water. Gel particles produced by autotrophs may be more surface active and more prone to aggregation (Zhou et al., 1998). The larger particle combined with the ballast effect of E.huxleyi are more easily to sink out of the SML. This, to a certain extent, may explain that a decrease in the EF's of CSP and TEP after the addition of E.huxleyi seed. The observed changes after addition of the E. huxleyi seed culture indicates that variations of gel particles in the SML may also depend on the source of gels and gel precursors. Zhou, J., Mopper, K., and Passow, U.: The role of surface-active carbohydrates in the formation of transparent exopolymer particles by bubble adsorption of seawater, Limnol Oceanogr, 43, 1860-1871, 1998.

---

## Author Comment (AC2) · 30 Jan 2018

We much appreciate the referee's constructive and thoughtful comments. Below we have pasted in the entire review, and inserted our responses to the suggestions (indicated by two stars).
detailed analysis of TEP and CSP concentration and size distribution. They concluded that wind speed controlled gel accumulation and size distribution in the SML under their experimental conditions. It is very difficult for me to evaluate the results and the conclusions in the present version of the manuscript because the description of the experiments as well as the presentation of some of the results are lacking important information (see details below). The manuscript would also benefit from shortening some of descriptive text in the Intro and in the Discussion. See below for some suggestions on that as well. I strongly recommend that the authors edit the text so that it is more focused and less wordy. ** We agree and will shorten the intro and discussion section.

1. Abstract: -In L. 9 use SML instead of surface microlayer. ** Will be done.

- Starting at L. 11: be more specific about the results on TEP and CSP; does this description refer to PSD of gels in the SML or bulk water? I suggest the following abbreviations for TEP and CSP in the SML (TEP-SML – CSP-SML) and in bulk water (TEP-bulk – CSP-bulk). Otherwise it is hard to distinguish between the two phases. ** This description refers to the PSD of gel particles in the SML. We think your suggestion is valuable. The abbreviations for TEP and CSP in the SML (TEP-SML – CSP-SML) and in bulk water (TEP-bulk – CSP-bulk) will be used across the whole MS to distinguish between the two phases.

- L. 17-18: You talk about the effects of TEP on aggregation and export. Since the focus of this paper is on TEP and CSP in the SML and the potential effects of gas exchange etc. you should focus/discuss potential effects on processes between the water and the atmosphere. In other words: if TEP settles out of the SML what that could mean for gas exchange processes between the water and the atmosphere. **With respect to the potential effects of the accumulation and size distribution of gels particles on the sea-air exchange process, the more detailed analysis on the fraction of submicron gel particles (0.4-1$\mu$m) will be addressed in the whole manuscript: Below information will be added in the abstract: The contribution of submicron gels particle in the smallest

size class, size 0.4-1$\mu$m, became larger at higher wind >6ms-1 after the addition of E. huxleyi, potentially impacting the emission of gels with sea spray aerosol.

The description below will be inserted in results section: 'The abundance fractions of submicron particles (0.4-1$\mu$m) in the SML were analyzed at low wind (LW) and high wind (HW) (Fig. S1). The results showed that the fraction of submicron gel particles became larger at high speed than at lower wind speed (<6.1 ms-1) during the period after addition of E. huxleyi followed by a biogenic SML from a previous experiment (p=0.003 for TEP-SML, p=0.02 for CSP-SML, two sample-Kolmogorov-Smirnov test). The median of fraction of submicron gel increased from 33.7% at low wind to 43.0% at high wind speeds for submicron TEP-SML and from 38.5% to 46.0% for submicron CSP-SML, respectively. However, there were no enhancement found in submicron fraction at high wind speed before the addition of E. huxleyi, with the exception on day11 for TEP when the fraction of submicron TEP-SML increased from 37.7% at 3.925ms-1 to 51.4% at 18.208 ms-1'.

The discussion below will be added: 'In this study, we found that the fraction of submicron gels (0.4-1$\mu$m) in the SML increased at high wind speeds after the addition of E. huxleyi and on day 11 with the peak concentration of bacterial abundance in SML. Due to the TEP's quasi-particulate nature, a considerable number of small gels can pass through a filter with size of 0.4$\mu$m (Passow and Alldredge,1995). It is therefore likely that the fraction of submicron gels was even higher at high wind speeds than observed. The changes of PSD in SML indicated that large gels were fragmented into smaller gels at high wind speed, or that submicron gels were generated. A strong enrichment of TEP in submicron SSA under field conditions has been observed before (Aller, et al 2017). Production of SSA in the field is driven by wind speed, and SSA in the size range 0.4-1$\mu$m in particular were observed to be higher at high wind speed (Lehahn et al., 2014).

References: Aller, J. Y., Radway, J. C., Kilthau, W. P., Bothe, D. W., Wilson, T. W., Vaillancourt, R. D., Quinn, P. K., Coffman, D. J., Murray, B. J., and Knopf, D. A.: Sizeresolved characterization of the polysaccharidic and proteinaceous components of sea spray aerosol, Atmos Environ, 154, 331-347, 2017. Lehahn, Y., Koren, I., Rudich, Y., Bidle, K. D., Trainic, M., Flores, J. M., Sharoni, S., and Vardi, A.: Decoupling atmospheric and oceanic factors affecting aerosol loading over a cluster of mesoscale North Atlantic eddies, Geophys Res Lett, 41, 4075-4081, 2014. Passow, U., and A. L. Alldredge, Aggregation of a diatom bloom in a mesocosm: The role of transparent exopolymer particles (TEP), Deep Sea Res., Part II, 42(1), 99–109, 1995.

2. Introduction: Page 3 - L. 6: I don't think you need the abbreviation ULW. ** We agree and will delete the abbreviation ULW.

- L. 9: do you have a reference for this statement? **References for this statement will be added:

Azetsu-Scott, K., and Niven, S. E. H.: The role of transparent exopolymer particles (TEP) in the transport of Th-234 in coastal water during a spring bloom, Cont Shelf Res, 25, 1133-1141, 10.1016/j.csr.2004.12.013, 2005. Ebling, A. M., and Landing, W. M.: Sampling and analysis of the sea surface microlayer for dissolved and particulate trace elements, Mar Chem, 177, 134-142, 10.1016/j.marchem.2015.03.012, 2015. Guasco, T. L., Cuadra-Rodriguez, L. A., Pedler, B. E., Ault, A. P., Collins, D. B., Zhao, D. F., Kim, M. J., Ruppel, M. J., Wilson, S. C., Pomeroy, R. S., Grassian, V. H., Azam, F., Bertram, T. H., and Prather, K. A.: Transition Metal Associations with Primary Biological Particles in Sea Spray Aerosol Generated in a Wave Channel, Environ Sci Technol, 48, 1324-1333, 10.1021/es403203d, 2014. Mari, X., Passow, U., Migon, C., Burd, A. B., and Legendre, L.: Transparent exopolymer particles: Effects on carbon cycling in the ocean, Prog Oceanogr, 151, 13-37, http://dx.doi.org/10.1016/j.pocean.2016.11.002, 2017.

- L. 14 -l. 2 on page 3 : In general, this text can be shortened as the focus is on SML sea-air exchange and not aggregation and particle export. Page 4 - L. 3: to me, your intro starts here. ** We agree and will shorten the Intro part.

- L. 25 – l. 4 on page 5: In the first sentence you are saying that "TEP enrichment : : : is inversely related to wind speed : : :". You don't have to repeat this statement in the following sentence; the first part of that sentence can be shortened: "One explanation for this is that : : :.". ** We agree and will delete the repeated statement according to your suggestion.

- L. what are the "other mechanisms" **It is proposed that gel particles formation within the SML is supported by bubble scavenging of DOM in the upper water column (Wurl et al., 2011), because more TEP precursors are lifted up the water-column. Moreover, compression and dilatation of the SML due to capillary waves may increase the rate of polymer collision, subsequently facilitating gel aggregation (Carlson, 1987).

Reference: Wurl, O., Wurl, E., Miller, L., Johnson, K., and Vagle, S.: Formation and global distribution of sea-surface microlayers, Biogeosciences, 8, 121-135, 10.5194/bg-8-121-2011, 2011. Carlson, D. J.: Viscosity of Sea-Surface Slicks, Nature, 329, 823-825, Doi 10.1038/329823a0, 1987.

3.Methods: Page 7 - L. 4: change to "November 3-24, 2014." **It will be done.

- L. 5: I am confused about the total volume of water collected for this study: Is it 20000 L with 14000 L of high sal water (what does high sal water mean??) + 8000 L at 5 m near Sylt? That does not add up, so remove "In total" in line 4, because your total is 42000 L. - L. 5: change to "were collected onboard FS Poseidon". How did you collect the water? Pumping or niskins? **20000 L is typo. It should be 22000L. The detail of sampling and collection will be presented in the method section:

'Effects of different wind speeds on the size distribution of organic gel particles in the SML were studied during the Aeolotron experiment from November 3-28, 2014. 22,000 L of North Atlantic seawater were collected by the research vessel POSEIDON, including ~14000L collected at 55 m at 64° 4,90' N, 8° 2,03' E and ~8000 L collected at 5 m depth near the Island of Sylt in the German Bight, North Sea. The water was pumped into a clean ("food save") road tanker and unloaded at the wind wave facility Aeolotron

in Heidelberg the following day and stored in the dark and cool (∼10°C) until the start of the experiment.'

L. 11-12: Info about something that you haven't used in your study like Uref is not important, so delete this sentence. ** We agree and will delete this part.

- L. 16: This is the part where I am getting confused about the experiments: 7 experiments were conducted, and you refer to fig.1 and table 1 for explanation. Figure 1 shows the step wise increase of U which lets me believe that the 7 experiments were conducted under the same conditions of U. Table 1 leads me with a different impression as the values of U were quite different throughout the experiments (the table is lacking the unit for U; you also need to describe what 'NaN' means. This needs to be explained in the methods. - L. 22-24: does this apply to all the 7 experiments? ** We agree that some description on the wind speed setting were confusing. More details about experiment will be presented in the revised version. The unit for U10 is m s-1 and it will be added in the table 1. NaN means no wind speed data on this condition. Information on the wind speed settingwill be added:

Two strategies of experimental wind speed setting were conducted in the experiment. For the first strategy, the wind speed setting was shown in the Figure1. 7 experiments were conducted on days 2, 4, 9, 11, 15, 22 and 24, respectively, with stepwise increase in wind speeds (equivalent to U10,) ranging from 1.371 to over 18.652 m s-1 as shown in Table 1. During some of the high wind speed conditions (Table 1), bubbles were generated in addition with a profiO2 oxygen diffuser hose to simulate strong breaking waves with bubble entrainment and spray formation. The second strategy was conducted on days 5, 12 and 23. Only one wind speed was arranged at about 18ms-1 with and without bubbling for about 2hour, respectively. The aim was to evaluate the difference effects between bubbling and no bubbling condition.

Why are there no values for U at some days during experiment 7. **U10 was determined by the method of Bopp and Jähne (2014). In this method, water velocity was

one of the important parameters to calculate the U10. Since data of water velocity at some conditions were absent, there no values for U10 could be obtained.

Page 8 - L.1-4: why was the light switched on in these two periods? Does that mean it was dark (0 umol m-2 s-1) throughout the rest of the incubation time? Why is this important? - L. 6: I could not find the Engel et al. 2017 reference in the list? Do you mean the Engel et al. (subm) reference? There is no way that we can get any information from this paper at this point. So you need delete this reference and give as many information of the methods as needed for this manuscript. - L. 9-11: why was E. hux added to the water? I suggest adding some explanation in the intro. Also, what do you mean by "adding a biogenic SML from a previous experiment"? That is too vague, I have no idea what a biogenic SML could be/look like, and how can this be added without disruption etc. - L. 19: It would help to show the collection volumes or give a range because it is hard to imagine how much water you collected from the SML. ** We will add details on the manipulations in the supplementary materials: During the experiment, a series of manipulations were conducted. To stimulate phytoplankton growth, lights were switched on from day 9 to day 16 and from day 20 to day 26, with a 12 Light:12 Dark regime, respectively,. On 14 November (day12), nutrients were added to final concentrations of 14.7 $\mu$mol L-1 nitrate (NO3), 9.5. $\mu$mol L-1 silicate (SiO4) and of 0.48 $\mu$mol L-1 phosphate (PO4). In order to induce phytoplankton growth and exudation, ~1L of an algal culture (Emiliania huxleyi , 4.6 x 105 cells ml-1) was added to the tank on day 20. In addition, 6L of water enriched with organic matter, sampled from surface microlayer during a previous phytoplankton mesocosm experiment, was added to the tank on day 21.This water had been stored frozen at -20° for about 6 month until the addition.

- L. 7-8: This statement is too general, and I don't see why this would be important to know at this point. **We agree with you and it will be deleted.

Page 10: - L.2: what are the wind conditions 1 and 2? **Wind conditions 1 and 2 were the first wind speed (1.66 ms-1) and the second wind speed (2.89 ms-1) condition on

day11.

4. Results: As mentioned above, I cannot evaluate the quality of the results before the authors improve the description of the experimental set-up. For example, I really cannot tell if the TEP and CSP results described on page 11 and shown in figure 2 are average values of all 7 experiments. Figure 2 also lacks error bars. You also need to add more detail to the figure legends (e.g. figs 4 and 5 show error bars, this needs to be mentioned in the legends). l. 16: this is the first time that chl a is mentioned. This needs to be described in the methods section. **Figure 2 showed average values of the different wind speed condition on each experimental day; the SD bars will be added. The error bars on figs 4 and 5 will be mentioned in the legends. The description on Chl a will be added in the method: 'Primary productivity was low during the whole experiment. Chlorophyll a (Chl a) concentrations were not detectable until days 20/21, after addition of the E. huxleyi culture and the SML water from a previous phytoplankton bloom experiment. Chl a concentration clearly increased after day 23'.

L. 20: what do you mean by "at the start of each wind experiment"?? Does that mean that you varied the wind speed over a course of a day from 0 - 20 or so (see also figures 4 and 5). **It means "at the start of experiment on each day of days 2, 4, 9, 11, 15, 22 and 24". On these experimental days, wind started at about 8:00 in the morning and ended at about 20:30 in the evening. The wind speeds over the seven experiment days varied a little, but all followed the same strategy of setting shown in the figure 1.

At last, we are thankful for your time and valuable suggestions to improve this manuscript.

[Figure]

**Fig. 1.** Figure S1 Changes in the submicron gel particles fraction with wind speed

---

## Author Response (AR1)

Dear editor,

We would like to thank the reviewers for giving us constructive suggestions which helped us to improve our manuscript. Here, we submit a thoroughly revised version and marked-up version of our manuscript, which has been modified according to the reviewers' suggestions. Efforts were also made to correct typos and to adjust decimal places of numbers of wind speed in the manuscript.

Below we have pasted in the entire review (bold font), and we have inserted our responses to the suggestions (indicated by bracketing stars).

**1. Response to the comments by referee#1**

(1) **Abstract:**

**Lines 2-9 I think this information can be abbreviated**

**We agree it, and L2-9, "The accumulation of gel…… ,so far, there is little ..." was deleted.*

**Line 15: : : and CSP? Complete the sentence.**

**The description on CSP slope changes was added according to the suggestion from referee#1. Please see below ( also please see p2, L10-13 in the revised version):*

*" The response of the $CSP_{SML}$ slopes to the wind speed varied through time of the experiment depending on the biogenetic source of gels. Wind speeds $>8ms^{-1}$ can decrease the slope of $CSP_{SML}$ significantly toward smaller size in the absence of*

*autotrophs condition."*

**(2) Introduction:**

  **I think it is too long and repetitive. Maybe the intro could be abbreviated and reorganized as follows: (1) introduce the SML and its properties. (2) introduce gels, their PSD, their biochemical relevance, and their accumulation and role in the SML (3) role of wind speed in SML formation and in gel dynamics, particularly in PSD**

*\*\* We agree and shortened the introduction into three parts: (1) introduction of gels and their role in SML, (2) PSD and their biochemical relevance, (3) role of wind speed in SML formation and in gel dynamics, including in PSD.*

*p2, L2-6, the sentences "The sea–surface microlayer (SML) is the thin…SML properties often differ from the underlying waters (ULW)" were deleted.*

*p2,L14-p3, L2, "TEP are sticky and can increase coagulation efficiencies of…. at the air-sea interface by contributing exudates and proteins released through cell disruption (Galgani and Engel, 2013)" were deleted.*

**Page 3 line 6-8 These sentences are repeating the same information.**

*\*\* The repeating information " have shown that marine organic gels can been riched in the SML, and" has been deleted. please see p3 L8-12 in the revised sentence.*

**Page 3 line 9: Start new line, since you talk about something different**

*\*\* It has been reorganized here. The introduction started here.*

**(3)  M&M:**

**Page 7 lines 7-9, how long did it take from sampling to start the experiment?**

*\*\*It took 41 days from sampling to start the experiment. The time of collection and starting experiment were added, please see p5 L23-p6 L6 in the revised version.*

*" Effects of different wind speeds on the size distribution of organic gel particles in the SML were studied during the Aeolotron experiment from November 3-28, 2014. 22,000 L of North Atlantic seawater were pumped and collected by the research vessel POSEIDON, including ~14000L collected at 55 m at 64°4.90' N, 8°2.03' E and ~8000 L collected on the 22. 09.2014 at 5 m depth near the Island of Sylt in the German Bight, North Sea. The water was pumped into a clean ("food save") road tanker and unloaded at the wind wave facility Aeolotron the following day and stored in the dark and cool (~10°C) until the start of the experiment. It took 41 days from sampling to start the experiment.    "*

**Page 9 lines 22-23 I do not think the calculation of TEP-C is necessary in this study that does not focus on carbon fluxes**

*\*\*We agree that this part has been removed.*

**Page 10 line 17 include space between 'distribution' and 'after'**

*\*\* It has been adopted.*

**(4) Results:**

**Page 11, lines 3-10 include average changes in TEP in the SML**

*\*\*We will add the average changes in TEP in the SML. Please see p10,L5-6 in the revised version.*

*" During the first two weeks, abundance and total area of $TEP_{SML}$ declined. After addition of the E. huxleyi seed culture and of pre-collected biogenic SML on day 20, $TEP_{SML}$ re-accumulated."*

**Figure 2. Is this the average of the different wind speed conditions? Clarify. Include SD bars. I would use the same symbols for the same parameters; e.g., if columns are for total area (as they are in SML and bulk CSP), then use also columns for TEP total area. Anyway, I do not think it is necessary to show the TEP-C; as your paper is not focused on these measurements. Include panel letters ABCD.**

*\*\* Yes, this is the average o f different wind speed conditions. and the SD bars have been added. The style of Figure 2 was revised according to the suggestion, using the same symbols for the same parameters. TEP-C was deleted.*

**Figure 3 and 4 say if this is SML or bulk water.**

*\*\* Figure 3 and 4 refer to the SML. This information was added in the revised manuscript and figure captions.*

**Figure 6. Day 22 panel: Use the same color and symbol code as in Figure 5 and in the rest of panels.**

*\*\*It has been adopted.*

**Page 11 lines 11-16. Here include average changes in CSP in bulk water**

*\*\*The information on average changes in CSP in bulk water has added in p10 L15-18:*

*"$CSP_{Bulk}$ concentration started with $12.9\pm10.7\times10^{6}$ $L^{-1}$ in abundance and $0.5\pm0.04\times10^{2}mm^{2}$ $L^{-1}$ in total area respectively, and increased to the first peak on day 9 for abundance and on day 5 for total area, and then declined (Fig.2 C, D). After day12, $CSP_{Bulk}$ concentration increased steadily."*

**Page 11 lines 14-15 in bulk or in SML?**

*\*\* We added the information 'in SML and bulk water'. please see it in p10 L21.*

**Section 3.3 Authors do not say whether they are describing PSD's in the SML or in bulk water at any moment. Assuming that this is only SML, some wording about changes in PSD in the bulk water could help understand these differences and to infer gel dynamics in the whole system through time**

*\*\*We agree that the assignment to the SML or to the bulk water is missing in the*

*current version. We thoroughly revise the text to give this information. Information of*

*PSD's in bulk water was added, please see p12,L22-p13,L6 in the revised version:*

*"Size distribution of gel particles (dp: 2-16μm) in the bulk water also followed the*

*power law relationship of Eq. (2) (mean of $r^2$=0.99), varying between -3.48 and -1.94*

*(mean value: -2.56, SD: 0.49) for $TEP_{Bulk}$ and between -3.43 and -2.01 (mean*

*value:-2.50, SD: 0.42) for $CSP_{Bulk}$. For the slopes of size distribution in the bulk water,*

*no significant difference was observed between high and low wind speeds. However,*

*as observed for the SML, the slopes of both TEP and CSP in the bulk water were*

*higher after adding the seed culture of E. huxleyi on day 20 and a biogenic SML from*

*a previous experiment on day 21 (Fig.8) (p <0.05, two sample-Kolmogorov-Smirnov*

*test), i.e., the average slope of $CSP_{Bulk}$ in size of 2-16μm was -2.84 before and -2.15*

*after addition of the E. huxleyi culture.*

**Page 12 lines 16-21 include some wording about enrichment factors in the high wind speed treatments**

*\*\*The information has been added in the revised version, please see p11, L21-23.*

*" Although the median of EF's were significantly lower at speed wind >6 $ms^{-1}$ than at*

*wind speed 2-6 $ms^{-1}$ (p <0.05; two-sample Kolmogorov-Smirnov test) (Table 2) gel*

*particles were not always depleted in the SML at high wind speeds".*

**Page 12 lines 23-24 and page 13: Where, in the SML or in bulk? This differentiation should be clearly stated across the whole MS.**

*\*\*It is in the SML. "$TEP_{SML}$, $CSP_{SML}$, $TEP_{Bulk}$ and $CSP_{Bulk}$ "  were used across the whole MS.*

**Page 13 lines 1-19 Maybe include the different slope values in a Table, as in the Figure it is hard to see if the difference is in slope or in the intercept.**

*\*\* We agree and added the table in the supplementary materials.*

**(4) Discussion:**

**Page 14 lines 8-12 I don't think this sentence is necessary since you are not discussing any results.**

*\*\* We agree and deleted it.*

**Page 14 line 24 remain enriched**

*\*\* It has been adopted.*

**Page 15 line 4-page 16 line 2. This paragraph is very long and it is not clear how it is connected to the results obtained, which I think should be more carefully introduced in the discussion: For instance, do you refer to your measurements in the SML, in bulk water, or in both? And, according to Kepkay 1994, shear is a dominant mechanism for particle aggregation; so how do you link this with the trend towards smaller gel particles at high wind speeds?**

*\*\* We modified this paragraph as below (also please see p14,L15-p15,L8 in the*

*revised version):*

*" The contribution of fraction of submicron gels particles became increasing when wind speed was above 6ms$^{-1}$, but the threshold of significant changing PSD in SML was wind speed of 8 ms$^{-1}$. Thus there is inharmonic effect of wind speeds on the submicron fraction and PSD. For higher wind speeds of 8 ms$^{-1}$ and above , the enhancement of shear and of kinetic energy dissipation by the release of momentum from the wave breaking (Donelan, 2013) were sufficiently energetic to bring about surface disruption and could result in more break-up of gel aggregates and changing PSD of gel particles. It should be noted that it need to set up the more experiments under the conditions of wind speed between 6ms$^{-1}$ and 8ms$^{-1}$ in the further study. Our results on the impact of wind speed on gel particles PSD corroborates earlier findings of Mari and Robert (2008). Aggregation processes are primarily driven by collision rates between particles that depend on particles concentration and turbulent shear (Ellis et al., 2004;Mccave, 1984;Mari and Robert, 2008). It has been suggested that TEP volume concentration increases continuously under the low turbulence intensity by promoting the formation of TEP, but that TEP volume concentration and the fraction of large TEP are reduced at stronger shear (Mari and Robert, 2008). Thus, the effect of wind shear on gel aggregation is double-edged, and large aggregates may be broken apart when the turbulence intensity increases. Our study suggests that high wind speed leads to a break-up of larger gel particles, enhancing the fraction of submicron gels in the SML."*

*References:*

*Ellis, K.M., Bowers, D.G., Jones, S.E.: A study of the temporal variability in particle size in a high energy regime. Coast. Shelf Sci. 61, 311–315,2004.*

*Mari, X., Robert, M.: Metal induced variations of TEP sticking properties in the southwestern lagoon of New Caledonia, Marine Chemistry, 110, 98–108, 2008.*

*Mccave, I. N.: Size Spectra and Aggregation of Suspended Particles in the Deep Ocean, Deep-Sea Res, 31, 329-352, Doi 10.1016/0198-0149(84)90088-8, 1984.*

**Page 16 lines 18-23. I do not see why. Average PSD are similar for TEP and CSP, and even lower for CSP at high wind speeds (page 13 lines 4-11). Or you said that because the change in PSD between high and low wind speeds was higher for TEP? Please clarify; and please refer to the results. To support this conclusion, maybe authors could look at the change in PSD of TEP and CSP through time; so check if these gel particles had been actually aggregated in the SML or not.**

*\*\* We modified this paragraph . Please see p15,L23-p16,L6*

"*According to our results, the average slopes showed about 41.2% changes for $TEP_{SML}$ at speed $> 8$ $ms^{-1}$ compared to low wind speed, but only 23.8% for $CSP_{SML}$. The change in slope of size distribution between high and low wind speeds was thus higher for $TEP_{SML}$ than $CSP_{SML}$. In addition, after adding the E. huxleyi seed culture, no influence of wind speed on size distribution of $CSP_{SML}$ was detected. These results indicated that the influence of wind speed on size distribution of gel particles may be more pronounced for $TEP_{SML}$ than for $CSP_{SML}$, and that $CSP_{SML}$ are less prone to*

*aggregation than TEP$_{SML}$ during the low wind speed. "*

**Section 4.3. I think it would be nice to comment about the changes in EF's through time. They apparently decrease until the phytoplankton culture is added (Table 2), even though you say that "a strong accumulation occurred in the SML (e.g. abstract line 13). How do you explain these decreases at low wind speeds? I would appreciate some comments about your day 15; any explanation to this exceptional behavior?**

*\*\*We think your suggestion is valuable.*

*We added the information (please see p19,L7-15 in the revised version): "In addition, pronounced changes through time in gel size slope and EF's were observed after the addition of E.huxleyi seed culture. At that time, shallower slopes for CSP and TEP revealed a higher abundance of larger gel particles relative to smaller ones for both SML and bulk water. Gel particles produced by autotrophs may be more surface active and more prone to aggregation (Zhou et al., 1998). The larger particle combined with the ballast effect of E.huxleyi are more easily to sink out of the SML. This, to a certain extent, may explain that a decrease in the EF's of CSP and TEP after the addition of the E.huxleyi seed. The observed changes after addition of the E. huxleyi seed culture indicates that variations of gel particles in the SML may also depend on the source of gels and gel precursors."*

**gas exchange etc. you should focus/discuss potential effects on processes between the water and the atmosphere. In other words: if TEP settles out of the SML what that could mean for gas exchange processes between the water and the atmosphere.**

 *** With respect to the potential effects of the accumulation and size distribution of gels particles on the sea-air exchange process, the more detailed analysis on the fraction of submicron gel particles (0.4-1μm) were addressed in the whole manuscript:*

*Below information were added in the abstract (please see p2,L15-21 in the revised version.): "Changes in spectral slopes between high wind speed and low wind speed were higher for TEP$_{SML}$ than for CSP$_{SML}$, indicating the impact of wind speed on size distribution of gel particles in the SML may be more pronounced for TEP than for CSP, and that CSP$_{SML}$ are less prone to aggregation than TEP$_{SML}$ during the low wind speed. The contribution of submicron gels particles in the smallest size class, size 0.4-1μm were enhanced at higher wind >6ms$^{-1}$ after the addition of an E. huxleyi culture potentially impacting the emission of gels with sea spray aerosol ".*

*The description below was inserted in results section (please see p13,L7-16 in the revised version ): "The abundance of submicron gel particles (0.4-1μm) in the SML were analyzed at low wind (LW) and high wind (HW) (Fig. S1), respectively. The results showed that the fraction of submicron gel particles became larger at high speed (>6.1 ms$^{-1}$) during the period after addition of E. huxleyi followed by a biogenic SML from a previous experiment (p=0.003 for TEP$_{SML}$, p=0.02 for CSP$_{SML}$,*

*two sample-Kolmogorov-Smirnov test). The median fraction of submicron gel increased from 33.7% at low to 43.0% at high wind speed for $TEP_{SML}$ and from 38.5% to 46.0% for $CSP_{SML}$, respectively. There was no enhancement found in submicron fraction at high wind speed before the addition of E. huxleyi, with the exception of day11 when the fraction of submicron $TEP_{SML}$ increased from 37.7% at 3.93 $ms^{-1}$ to 51.4% at 18.2 $ms^{-1}$."*

*The discussion below was added in the discussion section (please see p18,L16-p19,L6 in the revised version ):"'In this study, we found that the fraction of submicron gels (0.4-1µm) in the SML increased at high wind speeds ($>6 ms^{-1}$) after the addition of E. huxleyi and on day 11 with the peak concentration of bacterial abundance in SML (Fig 8). Due to the TEP's flexible nature, small gels can pass through a filter with size of 0.4 µm (Passow and Alldredge, 1995) and thus may escape the measurement. It is therefore likely that the fraction of submicron gels was even higher at high wind speeds than observed. The changes of size distribution of gels in SML indicated that large gels were fragmented into smaller gels at high wind speed, or that submicron gels were generated. A strong enrichment of TEP in submicron SSA under field conditions has been observed by Aller et al. (2017). Production of SSA in the field is driven by wind speed, and SSA in the size range 0.4-1 µm in particular were observed to be higher at high wind speed (Lehahn et al., 2014). Therefore, our finding support the results of Aller et al (2017) and Lehahn et al (2014) and suggest that the enhanced contribution of submicron gels particle at higher wind $>6ms^{-1}$ after the addition of E. huxleyi, potentially impact the emission of gels with sea spray aerosol.).*

*\*\*It has been done.*

**- L. 5: I am confused about the total volume of water collected for this study: Is it 20000 L with 14000 L of high sal water (what does high sal water mean??) + 8000 L at 5 m near Sylt? That does not add up, so remove "In total" in line 4, because your total is 42000 L. - L. 5: change to "were collected onboard FS Poseidon". How did you collect the water? Pumping or niskins?**

*\*\*20000 L is typo. It should be 22000L. The detail of sampling and collection were presented in the method section (please see p5 L23-p6 L6 in the revised version):*

*" Effects of different wind speeds on the size distribution of organic gel particles in the SML were studied during the Aeolotron experiment from November 3-28, 2014. 22,000 L of North Atlantic seawater were pumped and collected by the research vessel POSEIDON, including ~14000L collected at 55 m at 64°4.90'N, 8°2.03'E and ~8000 L collected on the 22. 09.2014 at 5 m depth near the Island of Sylt in the German Bight, North Sea. The water was pumped into a clean ("food save") road tanker and unloaded at the wind wave facility Aeolotron the following day and stored in the dark and cool (~10°C) until the start of the experiment. It took 41 days from sampling to start the experiment. "*

**L. 11-12: Info about something that you haven't used in your study like Uref is not important, so delete this sentence.**

*\*\* We agree and deleted this part.*

**- L. 16: This is the part where I am getting confused about the experiments: 7 experiments were conducted, and you refer to fig.1 and table 1 for explanation. Figure 1 shows the step wise increase of U which lets me believe that the 7 experiments were conducted under the same conditions of U. Table 1 leads me with a different impression as the values of U were quite different throughout the experiments (the table is lacking the unit for U; you also need to describe what 'NaN' means. This needs to be explained in the methods. - L. 22-24: does this apply to all the 7 experiments?**

*\*\* We agree that some description on the wind speed setting were confusing. More details about experiment were presented in the revised version. The unit for $U_{10}$ is m s$^{-1}$ and it will be added in the table 1. NaN means no wind speed data on this condition.*

*Information on the wind speed setting was added (please see p6 L11-25 in the revised version):*

*"On November 3rd 2014 (day1) the experiment started. Two strategies of experimental wind speed setting were applied during the experiment. For the strategy I, 7 experiments were conducted on days 2, 4, 9, 11, 15, 22 and 24, respectively, with stepwise increase in wind speeds (equivalent to $U_{10}$,) ranging from 1.37 to over 18.7 m s$^{-1}$ as shown in Table 1. At some conditions, data of water velocity were absent, hence no values for $U_{10}$ could be obtained. On experimental days, wind started at about 8:00 in the morning and ended at about 20:30 in the evening. The wind speeds*

*over the seven experiment days varied a little, but all followed the same strategy of setting shown in the conceptual figure 1. During some of the high wind speed conditions (Table 1), bubbles were generated in addition with a profiO$_2$ oxygen diffuser hose to simulate strong breaking waves with bubble entrainment and spray formation. Strategy II was followed on days 5, 12 and 23. Here, only one wind speed was applied (~18 ms$^{-1}$) with and without bubbling for about 2 hour, respectively. The aim was to evaluate the difference effects between bubbling and no bubbling condition. Seawater temperature over the course of the experiment was about 21± 1°C."*

**Why are there no values for U at some days during experiment 7.**

*\*\*U$_{10}$ was determined by the method of Bopp and Jähne (2014). In this method, water velocity was one of the important parameters to calculate the U$_{10}$. Since data of water velocity at some conditions were absent, there no values for U$_{10}$ could be obtained.*

**Page 8 - L.1-4: why was the light switched on in these two periods? Does that mean it was dark (0 umol m-2 s-1) throughout the rest of the incubation time? Why is this important? - L. 6: I could not find the Engel et al. 2017 reference in the list? Do you mean the Engel et al. (subm) reference? There is no way that we can get any information from this paper at this point. So you need delete this reference and give as many information of the methods as needed for this manuscript. - L. 9-11: why was E. hux added to the water? I suggest adding some**

**explanation in the intro. Also, what do you mean by "adding a biogenic SML from a previous experiment"? That is too vague, I have no idea what a biogenic SML could be/look like, and how can this be added without disruption etc. - L. 19: It would help to show the collection volumes or give a range because it is hard to imagine how much water you collected from the SML.**

*\*\* We added details on the manipulations in the supplementary materials:*

*" During the experiment, a series of manipulations were conducted. To stimulate phytoplankton growth, lights were switched on from day 9 to day 16 and from day 20 to day 26, with a 12 Light:12 Dark regime, respectively,. On 14 November (day12), nutrients were added to final concentrations of 14.7 µmol $L^{-1}$ nitrate ($NO_3$), 9.5. µmol $L^{-1}$ silicate ($SiO_4$) and of 0.48 µmol $L^{-1}$ phosphate ($PO_4$). In order to induce phytoplankton growth and exudation, ~1L of an algal culture (Emiliania huxleyi , 4.6 x $10^5$ cells $ml^{-1}$) was added to the tank on day 20. In addition, 6L of water enriched with organic matter, sampled from surface microlayer during a previous phytoplankton mesocosm experiment, was added to the tank on day 21.This water had been stored frozen at -20° for about 6 month until the addition."*

**- L. 7-8: This statement is too general, and I don't see why this would be important to know at this point.**

*\*\*We agree with you and it was deleted.*

**Page 10: - L.2: what are the wind conditions 1 and 2?**

*\*\*Wind conditions 1 and 2 were the first wind speed (1.66 ms$^{-1}$) and the second wind speed (2.89 ms$^{-1}$) condition on day11. please see p8,L22 in the revised version.*

**(4)Results:**

**As mentioned above, I cannot evaluate the quality of the results before the authors improve the description of the experimental set-up. For example, I really cannot tell if the TEP and CSP results described on page 11 and shown in figure 2 are average values of all 7 experiments. Figure 2 also lacks error bars. You also need to add more detail to the figure legends (e.g. figs 4 and 5 show error bars, this needs to be mentioned in the legends). l. 16: this is the first time that chl a is mentioned. This needs to be described in the methods section.**

*\*\*Figure 2 showed average values of the different wind speed condition on each experimental day; the SD bars were added.*

*The error bars on figs 4 and 5 were mentioned in the legends.*

*The description on Chl a was added in the method(please see p7 L9-12 in the revised version):*

*'Primary productivity was low during the whole experiment. Chlorophyll a (Chl a) concentrations were not detectable until days 20/21, after addition of the E. huxleyi culture and the SML water from a previous phytoplankton bloom experiment. Chl a concentration clearly increased after day 23'.*

**L. 20: what do you mean by "at the start of each wind experiment"?? Does that**

**mean that you varied the wind speed over a course of a day from 0 - 20 or so (see also figures 4 and 5).**

*\*\*It means "at the start of experiment on each day of days 2, 4, 9, 11, 15, 22 and 24".*

*On these experimental days, wind started at about 8:00 in the morning and ended at about 20:30 in the evening. The wind speeds over the seven experiment days varied a little, but all followed the same strategy of setting shown in the figure 1.*

***At last, we are thankful for your time and valuable suggestions to improve this manuscript.***

[revised manuscript text omitted]

Figure 7, A-D: Maximum size (ESD) of gel particles in the SML; A) and C): before addition of *E.huxleyi*; B) and D): after addition of *E.huxleyi*.

Figure 8: Average slopes of gel particles in the bulk water and SML. Open bars: before addition of *E.huxleyi,* hatched bars: after addition of *E.huxleyi,* error bars indicate ±1 SD.

Figure 9, A-G:  Strong accumulation of TEP and CSP in the SML at low wind speed as determined by microscopy, A: TEP (2.0 ms$^{-1}$), B: TEP (4.3 ms$^{-1}$), C: TEP

(8.3 ms$^{-1}$), D: CSP (2.0 ms$^{-1}$), E: CSP (4.3 ms$^{-1}$), F: CSP (8.3 ms$^{-1}$); G: Proposed schematic for interactions between wind speed and gel particle coverage in the SML.

[Figure]

                        Figure 1

[Figure]

[Figure]

Figure 2

[Figure]

                          Figure 3

[Figure]

                          Figure 4

[Figure]

.

[Figure]

Figure 5

[Figure]

.

[Figure]

Figure 6

[Figure]

[Figure]

                                  Figure 7

[Figure]

[Figure]

Figure 8

[Figure]

[Figure]

[Figure]

[Figure]

Figure 9

---

## Referee Report (RR1)

The authors addressed most of my comments from the first round of revisions. I now have a much better understanding of the wave tank experiments and the presentation of the results which, in my opinion, can still be shortened. I suggest focusing the results on the wind experiments in which the wind was stepwise increased over time (strategy I) and delete those from strategy II as they do not add much to the discussion (strategy I has the bubbling effect as well).  I also strongly encourage the authors to add more detail on sample volume and possible replicates (see below and my comments in the first round of the revision).

Methods:
p. 6: I do not think that the results from the experiments in which the wind speed was kept constant (strategy II) add much to the conclusions of the paper, or do they? I suggest taking them out. If the results need to be kept in the paper, you need to justify in the methods why the two different strategies were chosen (strategy I vs II).

p.6, l. 23 – p.7, l.3: This text needs to be moved into the results section.

p.7, l.7 – l. 10: Same thing: results, not methods.

p.7 – Sampling

The authors need to add more info on the quantity of samples taken from the SML and bulk water (I already mentioned that in the first round of revisions). I still don't know how much water was taken with the glass plate each sampling. You mention the numbers of dips but not the volume. Was the water from the 25 dips pooled into one sample or were those replicate samples? How about the bulk water samples? How much water was taken out? Replicates?

p. 8 -  Analytical methods

How many replicate filters were prepared for the TEP and CSP analysis?

p. 9, l. 17: Average values of n = ?

Results:

Text on page 10: these results are not necessary and should be taken out. They do not add anything to the conclusions of the manuscript. Focus your manuscript on the results of the 7 wind experiments (i.e. days 2, 4, 9, 11, 15, 22, 24 - strategy I).

Discussion

p. 14, l.6 – 11: this was already mentioned in the intro – shorten the text here.

p. 16, l. 6 – 10: this info is mainly introduction – needs to be shortened.

p. 18: l. 11 – 14: as above, needs to be shortened.

Typos/wording:

 p. 2, l. 9-10: change to 'smaller sizes'

l. 10-11: delete 'of the experiment'

l. 17: change to 'during low wind speeds.'

p. 3, l. 25: delete both 'the'

p. 4, l.2: 'Gel PSD …', new line

l. 14: change 'the' to 'a'

l. 19: 'TEP enrichment …', new line

p. 5, l. 15-17: delete this sentence

l. 23: change to '… Poseidon: ~14000 L …'

p. 6, l. 3: change to 'Aeolotron (Heidelberg, Germany)'

p. 9, l. 14: add space between 'distribution' and 'after'

l. 15: delete 'from a previous experiment'

l. 16-17: change to 'assuming a normal distribution of the data.'

p. 11, l. 8: change 'excluding' to 'except for' (and hereafter)

l. 21: change 'speed wind' to 'wind speed'

p. 12, l. 9: change 'identified also as' to 'the'

p.19, l. 2: change to 'gel particles'

---

## Author Response (AR2)

Dear Editor,

We much appreciate this referee's constructive and thoughtful comments. Below we have pasted in the entire review (bold font), and we have inserted our responses to the suggestions (indicated by bracketing stars).

**Editor's comments:**

**I have now received the 2 reviewers' comments on your revised manuscript. As you can see, both reviewers agree that the manuscript has been substantially improved and that most of their comments have been incorporated. At the same time, however, they also still detected a number of typos and more importantly, suggest to make some alterations on the revised manuscript. One reviewer suggests, for example, to delete the description, results, etc of one experiment where the wind speed was kept constant.**

*\*\* We agree with all suggestions from referees, and deleted the description, results and discussion where the wind speed was kept constant accordingly. Typos have been corrected in the manuscript.*

**Referee 1:**

**Suggestions for revision or reasons for rejection (will be published if the paper is accepted for final publication)**
**I thank the authors for their careful revision of the MS, that looks improved in its present form. I still have some comments to make that in my opinion should be addressed:**

**The introduction section looks now much clearer and the objectives of the paper are well presented. However, I would suggest to unify the term "gels", as you use many (e.g. biogenic gels, gel-like substances, microgels, etc)**

*\*\*It has been done. We unified the term "gels" throughout the manuscript.*

**M&M. What time was the SML and bulk water sampled? In the morning and at 20:30 evening? Or only in the evening?**

*\*\*The SML was taken at the end of each wind condition. A pair of SML and Bulk water samples was collected at each wind conditions except for day2 and day4. On day2 and day4, the bulk water was collected at the start(morning) and the end (evening) of the experiment. Compared to the significant changes of gel concentration in SML with wind speeds, the gel concentration changes smaller with wind speeds in bulk water (data not shown). Therefore, the average of gel concentration in bulk was not sensitive to wind speed changes.*

*The description on the time of water sampled were added to the M&M. Please see Line21, P6-Line3, P7 in the MS.*

**Figure 2: Does this correspond to the average of all experiments? With sampling done at the end of each one? Please clarify. Also, include TEP and CSP in the axis titles**

*\*\* "TEP" and "CSP" have been added to axis titles.*

*For SML samples, this corresponded to the average gels concentration of all wind speed conditions on each one experiment day. Sampling of SML was done at the end of each one wind speed condition.*

*Bulk water was sampled at the end of each one wind speed condition excepted for day2 and day4. On day2 and day4, bulk samples were collected at the first wind speed condition (morning) and the end wind speed condition(evening). Compared to the significant changes of gel concentration in SML with wind speeds, the gel concentration changes smaller with wind speeds in bulk water (data not shown). Therefore, the average of gel concentration in bulk was not sensitive to wind speed changes.*

*In addition, this section was moved to the supplementary materials according to the other referee's suggestion. Please see Line24-25, P6 in the MS.*

**Table 2: Why, if samples were taken at each wind level (see Figures 3 and 4), EF are not calculated for all of them? (e.g. days 2 and 11)**

\*\*On day2, *bulk samples were only collected at the first wind speed condition (morning) and the end wind speed condition(evening). On day 11, SML samples on condition 1 and condition 2 were contaminated. Therefore, EF were not calculated for all of them.*

**Discussion: I think too much importance in the discussion is given to the effect of bubbles, when no significant and consistent effect was observed (Table 3), I would rather conclude that your experiments did not help understand the effect of bubbles on SML enrichements and further work is neccesary.**

*\*\* We agree with the referee's comments. The results and discussion on this point were deleted, since the limited data from this study prohibited to reach the conclusion of the bubble effect on the gel enrichment in SML.*

**Referee 2:**
**The authors addressed most of my comments from the first round of revisions. I now have a much better understanding of the wave tank experiments and the presentation of the results which, in my opinion, can still be shortened. I suggest focusing the results on the wind experiments in which the wind was stepwise increased over time (strategy I) and delete those from strategy II as they do not add much to the discussion (strategy I has the bubbling effect as well). I also strongly encourage the authors to add more detail on sample volume and possible replicates (see below and my comments in the first round of the revision).**

*\*\* We agree with the referee's comments. The description on strategy II was deleted from the manuscript.*

**Methods:**

**p. 6: I do not think that the results from the experiments in which the wind speed was kept constant (strategy II) add much to the conclusions of the paper, or do they? I suggest taking them out. If the results need to be kept in the paper, you need to justify in the methods why the two different strategies were chosen (strategy I vs II).**

*\*\* The results and discussion on the bubble effect were deleted, since the limited data from this study prohibited to reach the conclusion of the bubble effect on the gel enrichment in SML.*

**p.6, l. 23 – p.7, l.3: This text needs to be moved into the results section.**

*\*\* It has been moved to results section.*

**p.7, l.7 – l. 10: Same thing: results, not methods.**

*\*\*It has been moved to results section.*

**p.7 – Sampling**

**The authors need to add more info on the quantity of samples taken from the SML and bulk water (I already mentioned that in the first round of revisions). I still don't know how much water was taken with the glass plate each sampling. You mention the numbers of dips but not the volume. Was the water from the 25 dips pooled into one sample or were those replicate samples? How about the bulk water samples? How much water was taken out? Replicates?**

*\*\*Sample volumes were 210-355ml for SML and 800-1000 ml for bulk water, respectively. It has been added to the sampling section in M&M. The water from all dips pooled into one sample. (Please see Line10, P7)*

**p. 8 - Analytical methods**

**How many replicate filters were prepared for the TEP and CSP analysis?**

*\*\*Two replicate filters were prepared for the TEP and CSP analysis. It has been clarified in the methods section (Please see Line2, P8).*

**p. 9, l. 17: Average values of n = ?**

*\*\* Here "Average values" are given by the statistical mean and its standard deviation (SD)".*

**Results:**

**Text on page 10: these results are not necessary and should be taken out. They do not add anything to the conclusions of the manuscript. Focus your manuscript on the results of the 7 wind experiments (i.e. days 2, 4, 9, 11, 15, 22, 24 - strategy I).**

*\*\*This section was moved to the supplementary materials.*

**Discussion**

**p. 14, l.6 – 11: this was already mentioned in the intro – shorten the text here.**

*\*\* It has been done.*

**p. 16, l. 6 – 10: this info is mainly introduction – needs to be shortened.**

*\*\* It has been done.*

**p. 18: l. 11 – 14: as above, needs to be shortened.**

*\*\* It has been done.*

**Typos/wording:**

**p. 2, l. 9-10: change to 'smaller sizes'**

**l. 10-11: delete 'of the experiment'**

**l. 17: change to 'during low wind speeds.'**

**p. 3, l. 25: delete both 'the'**

**p. 4, l.2: 'Gel PSD …', new line**

**l. 14: change 'the' to 'a'**

**l. 19: 'TEP enrichment …', new line**

**p. 5, l. 15-17: delete this sentence**

**l. 23: change to '… Poseidon: ~14000 L …'**

**p. 6, l. 3: change to 'Aeolotron (Heidelberg, Germany)'**

**p. 9, l. 14: add space between 'distribution' and 'after'**

**l. 15: delete 'from a previous experiment'**

**l. 16-17: change to 'assuming a normal distribution of the data.'**

**p. 11, l. 8: change 'excluding' to 'except for' (and hereafter)**

**l. 21: change 'speed wind' to 'wind speed'**

**p. 12, l. 9: change 'identified also as' to 'the'**

**p.19, l. 2: change to 'gel particles'**

*\*\* All typos and the wording have been revised accordingly.*

[revised manuscript text omitted]

3                              Figure 1

[Figure]

6          Figure 2

[Figure]

[Figure]

4                                  Figure 3

[Figure]

Figure 4

[Figure]

2          Figure 5

[Figure]

2                Figure 6

[Figure]

2 Figure 7

[Figure]

[Figure]

Figure 8